# CosNet: A Generalized Spectral Kernel Network

**Yanfang Xue**[1,2]**, Pengfei Fang**[1,2]**, Jinyue Tian**[1,2]**, Shipeng Zhu**[1,2]**, Hui Xue**[1,2]*
[1]School of Computer Science and Engineering, Southeast University, Nanjing, 210096, China
[2]Key Laboratory of New Generation Artificial Intelligence Technology and Its Interdisciplinary
Applications (Southeast University), Ministry of Education, China
{230218795, fangpengfei, 220222083, shipengzhu, hxue}@seu.edu.cn

## Abstract

Complex-valued representation exists inherently in the time-sequential data that can be derived from the integration of harmonic waves. The non-stationary spectral kernel, realizing a complex-valued feature mapping, has shown its potential to analyze the time-varying statistical characteristics of the time-sequential data, as a result of the modeling frequency parameters. However, most existing spectral kernel-based methods eliminate the imaginary part, thereby limiting the representation power of the spectral kernel. To tackle this issue, we propose a generalized spectral kernel network, namely, Complex-valued spectral kernel Network (Cos-Net), which includes spectral kernel mapping generalization (SKMG) module and complex-valued spectral kernel embedding (CSKE) module. Concretely, the SKMG module is devised to generalize the spectral kernel mapping in the real number domain to the complex number domain, recovering the inherent complex-valued representation for the real-valued data. Then a following CSKE module is further developed to combine the complex-valued spectral kernels and neural networks to effectively capture long-range or periodic relations of the data. Along with the CosNet, we study the effect of the complex-valued spectral kernel mapping via theoretically analyzing the bound of covering number and generalization error. Extensive experiments demonstrate that CosNet performs better than the mainstream kernel methods and complex-valued neural networks.

## 1 Introduction

Complex numbers represent the information of the amplitude and phase simultaneously. In contrast to the amplitude, which can be revealed by the real number, the phase can denote the time delay and advance, thereby encoding the temporal dependency of the data Hirose [2003]. This suggests that complex-valued models can be employed in some practical applications, especially for the time-sequential data where information is wave-related, such as signal analysis Hirose *et al.* [2019]; Yu *et al.* [2019]; Zeng *et al.* [2022], speech processing Shafran *et al.* [2018], and time series classification Yang *et al.* [2020, 2017]; Wisdom *et al.* [2016].

In the learning community, the spectral kernel, which is constructed from the inverse Fourier transform, can naturally realize the complex-valued mapping. That said, the spectral kernel can analyze the data in the frequency domain directly. As a candidate of the spectral kernel family, the non-stationary spectral kernel is also proposed to conquer the local limitations of the classical kernels, such as stationarity and monotonicity Remes *et al.* [2017]; Tompkins *et al.* [2020]; Ton *et al.* [2018]; Li *et al.* [2020]. Ideally, these advanced kernels can extract appropriate non-stationary time-varying characteristics of the data by modeling the frequency parameters, and hence infer the long-range or periodic relations of the input data.

---

*Corresponding author

37th Conference on Neural Information Processing Systems (NeurIPS 2023).

However, most existing methods usually eliminate the imaginary part of the spectral kernel mapping arbitrarily for the convenience of calculation. For example, Rahimi and Recht [2007]; Zhang *et al.* [2017a] ignore the imaginary part directly by replacing the integrand $e^{j\boldsymbol{\omega}(\boldsymbol{x}-\boldsymbol{x}')}$ with $\cos(\boldsymbol{\omega}(\boldsymbol{x}-\boldsymbol{x}'))$; Xue *et al.* eliminates the imaginary part with an elaborate spectral density function definition for the non-stationary spectral kernel Xue *et al.* [2019]. Remarkably, existing research shows that complex numbers could lead to a rich representational capability for wave-related information processing Wisdom *et al.* [2016]; Danihelka *et al.* [2016]; Worrall *et al.* [2017]; Trouillon and Nickel [2017]. However, simply plugging the imaginary part in the neural networks does not ensure that the model retains the property of the spectral kernel. Therefore, more efforts are required to develop a new framework that can involve the imaginary part in spectral kernel networks.

In this paper, we propose a new framework that generalizes the spectral kernel that endows with the complex-value representation, and we name it as a complex-valued spectral kernel network (CosNet). The proposed CosNet includes two modules: the spectral kernel mapping generalization (SKMG) module and the complex-valued spectral kernel embedding (CSKE) module. Technically, we generalize the spectral kernel mapping in the real number domain to the complex number domain by defining the spectral density function in the SKMG module. We further embed the complex-valued spectral kernel into neural networks to attain the proposed CosNet using the CSKE module. It is noted that a new initialization scheme is also proposed for the CSKE module that adopts the cosine and sine functions as the activation for the real and imaginary parts of the weight matrix. This initializing scheme retains the statistical characteristics of the non-stationary spectral kernel. It enables CosNet to take the relative distance of data into account by shifting between phases, such that capture the long-range or periodic relations of data in the complex domain without increasing the number of parameters. Our contributions in this paper are shown as follows:

- We propose a complex-valued spectral kernel network, i.e., CosNet, which takes both the real and imaginary parts of the spectral kernel mapping into account and thus improves the representational capability of the spectral kernel.

- We propose an initialization scheme for the complex-valued weight matrix, which ensures that CosNet retains the property of non-stationary spectral kernels and takes the relative distance of data in the complex number domain without increasing the number of parameters.

- We provide the lower generalization bound of CosNet than the real-valued non-stationary spectral kernel.

- Thorough experiments demonstrate that our proposed method is totally superior to state-of-the-art kernel methods.

## 2   Related Work

**Spectarl kernel networks**    Spectral approaches were developed to fully characterize general kernels with concise representation forms, such as sparse spectrum kernels Lázaro-Gredilla *et al.* [2010], sparse mixture kernels Wilson and Adams [2013], non-stationary spectral kernels Remes *et al.* [2017], and random Fourier features methods to deal with large-scale settings Li *et al.* [2019]; Liu *et al.* [2021]. These methods commonly approximated the kernel function using an explicit spectral representation based on Bochner's theorem Bochner and others [1959] and Yaglom's theorem Yaglom [1987]. Benefiting from the outstanding representation capability of neural networks with hierarchical nonlinear linking structures Bengio *et al.* [2006], researchers attempt to embed spectral representation (i.e., feature mapping of kernels) into the hierarchical architecture of neural networks to construct spectral kernel networks. Zhang *et al.* [2017a] used the Random Fourier Feature to approach the stationary kernel mapping and embedded it into each layer of DNNs. Xue *et al.* [2019] proposed a deep spectral kernel network to embed the non-stationary spectral kernel into each layer of DNNs, which can approximate most of the kernels. Li *et al.* [2020] proposed an automated spectral kernel learning (ASKL) that incorporates the process of finding suitable non-stationary kernels and model training. However, for convenience of calculation, these models commonly use the real-valued representation, although spectral kernels lead to the complex-valued mapping.

**Complex-valued neural networks**    Complex-valued neural networks (CVNNs) have shown excellent efficiency compared to their real counterparts in biological Reichert and Serre [2013], speech enhancement Tsuzuki *et al.* [2013]; Choi *et al.* [2019], image Popa [2017]; Wen *et al.* [2020], and signal processing Kim and Guest [1990]; Wilmanski *et al.* [2016]. In previous studies, researchers

commonly split the complex-valued input into a pair of real-valued inputs and fed them into the real-valued neural networks with both real-valued weight matrix and activation function. This design cannot exploit the advantages of complex numbers completely, and the neural network convergence strongly depends on proper initialization and the choice of learning rate Yang *et al.* [2007]; Zhang *et al.* [2009]. Subsequently, CVNNs with complex-valued weight and activation functions are proposed in the complex number domain to deal with complex-valued inputs Hirose [1992]; Dedmari *et al.* [2018]; Zhang *et al.* [2017b]. Benefitting from the rich representation capability, researchers tend to extend CVNNs to other neural networks, such as complex-valued convolutional neural networks Trabelsi *et al.* [2018], complex-valued residual neural networks Wang *et al.* [2018], and complex-valued recurrent neural networks Wolter and Yao [2018]; Arjovsky *et al.* [2016]. All these works have proved that the complex-valued models have a richer representational capacity and perform better on real-world learning tasks by a set of experiments.

## 3 Complex-valued Spectral Kernel Networks

In this section, we first introduce concepts and notations of the non-stationary kernel and complex numbers. Then, we provide the overall architecture of our CosNet with two modules. Moreover, we explicitly provide the details of each module. In addition, we present a detailed analysis of CosNet.

### 3.1 Preliminary

To better illustrate CosNet, we introduce the necessary preliminary knowledge and notation of non-stationary spectral kernels and complex numbers in this section.

**Notations**    Formally, we use $\mathbb{R}^n$, $\mathbb{C}^n$, $\mathbb{R}^{m \times n}$ and $\mathbb{C}^{m \times n}$ to denote $n$-dimensional Euclidean spaces, $n$-dimensional complex number spaces, the space of $m \times n$ real-valued matrix and the space of $m \times n$ complex-valued matrix. Throughout the paper, the matrices, vectors and scalars are denoted by bold capital letters (*e.g.* $\boldsymbol{X}$), bold lower-case letters (*e.g.* $\boldsymbol{x}$) and lower-case letters (*e.g.* $x$), respectively. A complex number $\boldsymbol{z} \in \mathbb{C}^D$ is represented as $\boldsymbol{z} = \boldsymbol{u} + i\boldsymbol{v}$ with a real part $\boldsymbol{u}$ and an imaginary part $\boldsymbol{v}$. $\bar{\boldsymbol{z}} = \boldsymbol{u} - i\boldsymbol{v}$ denotes the complex conjugate of $\boldsymbol{z}$. For any two complex numbers $\boldsymbol{z}_1 = \boldsymbol{u}_1 + i\boldsymbol{v}_1, \boldsymbol{z}_2 = \boldsymbol{u}_2 + i\boldsymbol{v}_2 \in \mathbb{C}$, $\boldsymbol{z}_1 + \boldsymbol{z}_2 = (\boldsymbol{u}_1 + \boldsymbol{u}_2) + i(\boldsymbol{v}_1 + \boldsymbol{v}_2)$, $\boldsymbol{z}_1 \boldsymbol{z}_2 = (\boldsymbol{u}_1 \boldsymbol{u}_2 - \boldsymbol{v}_1 \boldsymbol{v}_2) + i(\boldsymbol{u}_1 \boldsymbol{v}_2 + \boldsymbol{v}_1 \boldsymbol{u}_2)$. To represent the complex-valued layer with $2D$ features, we allocate the first $D$ features to represent the real component and the remaining to represent the imaginary component.

**Preliminary knowledge**    Non-stationary spectral kernels are constructed from inverse Fourier transform in the frequency domain. Based on Yaglom's theorem Yaglom [1987], a general kernel $k(\boldsymbol{x}, \boldsymbol{x}')$ is positive definite on $\mathbb{R}^D$ if and only if it admits the form:

$$k(\boldsymbol{x}, \boldsymbol{x}') = \int_{\mathbb{R}^D \times \mathbb{R}^D} e^{i(\boldsymbol{\omega}^\top \boldsymbol{x} - \boldsymbol{\omega}'^\top \boldsymbol{x}')} \mu(d\boldsymbol{\omega}, d\boldsymbol{\omega}') \tag{1}$$

where $\mu(d\boldsymbol{\omega}, d\boldsymbol{\omega}')$ is the Lebesgue-Stieltjes measure associated with some positive semi-definite spectral density function $s(\boldsymbol{\omega}, \boldsymbol{\omega}')$ with bounded variations. Therefore, a general kernel can be defined as the following form:

$$k(\boldsymbol{x}, \boldsymbol{x}') = \int_{\mathbb{R}^D \times \mathbb{R}^D} e^{i(\boldsymbol{\omega}^\top \boldsymbol{x} - \boldsymbol{\omega}'^\top \boldsymbol{x}')} s(\boldsymbol{\omega}, \boldsymbol{\omega}') d\boldsymbol{\omega} d\boldsymbol{\omega}' \tag{2}$$

where $s(\boldsymbol{\omega}, \boldsymbol{\omega}')$ can be understood as a joint probability density function.

### 3.2 Overall architecture

To explore the capability of the imaginary part in the spectral kernel networks, we propose CosNet as a generalized framework. CosNet involves two modules: the SKMG module to achieve complex-valued spectral kernel mapping and the CSKE module to embed the spectral kernel into neural networks. The overall architecture is shown in Figure 1.

Concretely, the SKMG module is denoted as $\Phi(\boldsymbol{x})$ via generalizing the spectral kernel mapping in the real number domain to the complex number domain for the real-valued data $\boldsymbol{x}$. And the CSKE module is denoted as $\Psi(\boldsymbol{h})$ via initializing complex-valued weight matrix with the cosine and sine function for the complex-valued spectral kernel mapping $\boldsymbol{h}$.

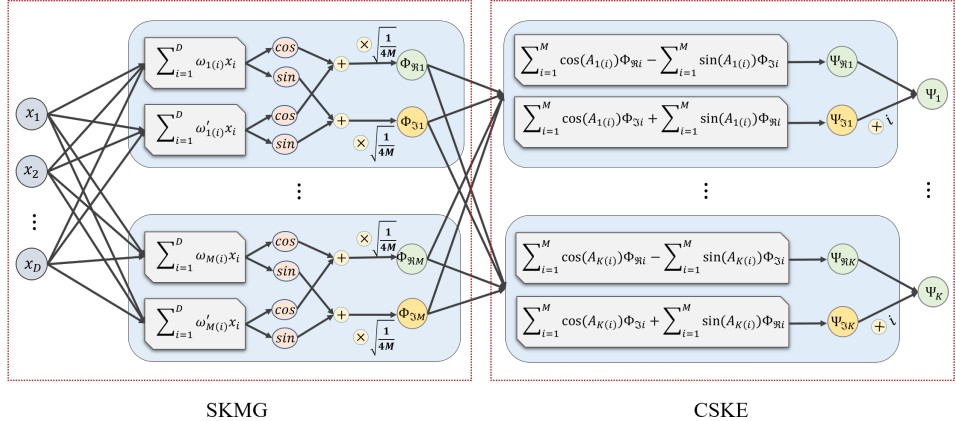

SKMG             CSKE

Figure 1: The structure of CosNet with two modules. The SKMG module is used to map the real-valued inputs to a complex-valued representation. The CSKE module is the complex-valued spectral kernel embedding with our initialization.

Based on the two modules, our CosNet with $l$ layers is defined as :

$$CosNet(\boldsymbol{x}) = \Psi^{l-1}(\ldots\Psi^1(\Phi^1(\boldsymbol{x}))). \tag{3}$$

Moreover, the corresponding complex-valued spectral kernel is defined as:

$$\mathbb{K}^{(l)}(\boldsymbol{x},\boldsymbol{x}') = \langle\Psi^{l-1}(\ldots\Psi^1(\Phi^1(\boldsymbol{x}))),\Psi^{l-1}(\ldots\Psi^1(\Phi^1(\boldsymbol{x}')))\rangle = \langle CosNet(\boldsymbol{x}),CosNet(\boldsymbol{x}')\rangle \tag{4}$$

where $\mathbb{K}^{(l)}(\boldsymbol{x},\boldsymbol{x}')$ denotes the $l$-layer complex-valued spectral kernel.

### 3.3 Complex-valued spectral kernel network (CosNet)

**Spectral kernel mapping generalization module** In this module, we generalize the spectral kernel mapping in the real number domain to the complex number domain. Furthermore, the generalized mapping can be used in both stationary and non-stationary spectral kernels. Here we elaborate on the detailed process.

According to Equation (2), to produce a positive semi-definite kernel, we need to include symmetries $s(\boldsymbol{\omega},\boldsymbol{\omega}') = s(\boldsymbol{\omega}',\boldsymbol{\omega})$ and sufficient diagonal components $s(\boldsymbol{\omega},\boldsymbol{\omega})$ and $s(\boldsymbol{\omega}',\boldsymbol{\omega}')$. Concretely, we replace the exponential component $e^{i(\boldsymbol{\omega}^\top\boldsymbol{x}-\boldsymbol{\omega}'^\top\boldsymbol{x}')}$ in Equation (2) with $\zeta_{\boldsymbol{\omega},\boldsymbol{\omega}'}(\boldsymbol{x},\boldsymbol{x}')$, which is defined as:

$$\zeta_{\boldsymbol{\omega},\boldsymbol{\omega}'}(\boldsymbol{x},\boldsymbol{x}') = \frac{1}{4}\Big[e^{i(\boldsymbol{\omega}^\top\boldsymbol{x}-\boldsymbol{\omega}'^\top\boldsymbol{x}')} + e^{i(\boldsymbol{\omega}'^\top\boldsymbol{x}-\boldsymbol{\omega}^\top\boldsymbol{x}')} + e^{i(\boldsymbol{\omega}^\top\boldsymbol{x}-\boldsymbol{\omega}^\top\boldsymbol{x}')} + e^{i(\boldsymbol{\omega}'^\top\boldsymbol{x}-\boldsymbol{\omega}'^\top\boldsymbol{x}')}\Big]. \tag{5}$$

Then, we expand the exponential component to the complex-valued representation with the cosine and sine function based on Euler's formula, and $\zeta_{\boldsymbol{\omega},\boldsymbol{\omega}'}(\boldsymbol{x},\boldsymbol{x}')$ can be rewritten as:

$$\begin{aligned}\zeta_{\boldsymbol{\omega},\boldsymbol{\omega}'}(\boldsymbol{x},\boldsymbol{x}') =\frac{1}{4}\Big[&\cos(\boldsymbol{\omega}^\top\boldsymbol{x}-\boldsymbol{\omega}'^\top\boldsymbol{x}')+i\sin(\boldsymbol{\omega}^\top\boldsymbol{x}-\boldsymbol{\omega}'^\top\boldsymbol{x}')\\ +&\cos(\boldsymbol{\omega}'^\top\boldsymbol{x}-\boldsymbol{\omega}^\top\boldsymbol{x}')+i\sin(\boldsymbol{\omega}'^\top\boldsymbol{x}-\boldsymbol{\omega}^\top\boldsymbol{x}')\\ +&\cos(\boldsymbol{\omega}^\top\boldsymbol{x}-\boldsymbol{\omega}^\top\boldsymbol{x}')+i\sin(\boldsymbol{\omega}^\top\boldsymbol{x}-\boldsymbol{\omega}^\top\boldsymbol{x}')\\ +&\cos(\boldsymbol{\omega}'^\top\boldsymbol{x}-\boldsymbol{\omega}'^\top\boldsymbol{x}')+i\sin(\boldsymbol{\omega}'^\top\boldsymbol{x}-\boldsymbol{\omega}'^\top\boldsymbol{x}')\Big].\end{aligned} \tag{6}$$

As a result, the general spectral kernel in Equation (2) can be redefined as:

$$k(\boldsymbol{x},\boldsymbol{x}') = \int_{\mathbb{R}^D\times\mathbb{R}^D}\zeta_{\boldsymbol{\omega},\boldsymbol{\omega}'}(\boldsymbol{x},\boldsymbol{x}')p(\boldsymbol{\omega},\boldsymbol{\omega}')d\boldsymbol{\omega}d\boldsymbol{\omega}' \tag{7}$$

where $p(\boldsymbol{\omega},\boldsymbol{\omega}') = \frac{1}{4}[s(\boldsymbol{\omega},\boldsymbol{\omega}') + s(\boldsymbol{\omega}',\boldsymbol{\omega}) + s(\boldsymbol{\omega},\boldsymbol{\omega}) + s(\boldsymbol{\omega}',\boldsymbol{\omega}')]$ also can be considered as a probability density function.

Subsequently, we approximate Equation (7) with Monte Carlo random sampling:

$$k(\boldsymbol{x}, \boldsymbol{x}') = \int_{\mathbb{R}^D \times \mathbb{R}^D} \zeta_{\boldsymbol{\omega},\boldsymbol{\omega}'}(\boldsymbol{x}, \boldsymbol{x}') p(\boldsymbol{\omega}, \boldsymbol{\omega}') d\boldsymbol{\omega} d\boldsymbol{\omega}' = \mathbb{E}_{\boldsymbol{\omega},\boldsymbol{\omega}' \sim P}\Big[\zeta_{\boldsymbol{\omega},\boldsymbol{\omega}'}(\boldsymbol{x}, \boldsymbol{x}')\Big]$$

$$\approx \frac{1}{4M} \sum_{i=1}^{M} \Big[\cos(\boldsymbol{\omega}_i^\top \boldsymbol{x} - \boldsymbol{\omega}_i'^\top \boldsymbol{x}') + i\sin(\boldsymbol{\omega}_i^\top \boldsymbol{x} - \boldsymbol{\omega}_i'^\top \boldsymbol{x}')$$
$$+ \cos(\boldsymbol{\omega}_i'^\top \boldsymbol{x} - \boldsymbol{\omega}_i^\top \boldsymbol{x}') + i\sin(\boldsymbol{\omega}_i'^\top \boldsymbol{x} - \boldsymbol{\omega}_i^\top \boldsymbol{x}')$$
$$+ \cos(\boldsymbol{\omega}_i^\top \boldsymbol{x} - \boldsymbol{\omega}_i^\top \boldsymbol{x}') + i\sin(\boldsymbol{\omega}_i^\top \boldsymbol{x} - \boldsymbol{\omega}_i^\top \boldsymbol{x}')$$
$$+ \cos(\boldsymbol{\omega}_i'^\top \boldsymbol{x} - \boldsymbol{\omega}_i'^\top \boldsymbol{x}') + i\sin(\boldsymbol{\omega}_i^\top \boldsymbol{x} - \boldsymbol{\omega}_i^\top \boldsymbol{x}')\Big]$$
$$= \langle \Phi(\boldsymbol{x}), \overline{\Phi(\boldsymbol{x}')} \rangle \tag{8}$$

where, $(\boldsymbol{\omega}_i, \boldsymbol{\omega}_i')_{i=1}^D$ is the frequency pairs, $M$ is the sampling number.

The generalized spectral kernel mapping in Equation (8) is defined as:

$$\Phi(\boldsymbol{x}) = \sqrt{\frac{1}{4M}} \Big[(\cos(\boldsymbol{\Omega}^\top \boldsymbol{x}) + \cos(\boldsymbol{\Omega}'^\top \boldsymbol{x})) + i(\sin(\boldsymbol{\Omega}^\top \boldsymbol{x}) + \sin(\boldsymbol{\Omega}'^\top \boldsymbol{x}))\Big], \tag{9}$$

and the frequency matrices $\boldsymbol{\Omega}, \boldsymbol{\Omega}'$ are denoted as:

$$\boldsymbol{\Omega} = [\boldsymbol{\omega}_1, \boldsymbol{\omega}_2, \cdots, \boldsymbol{\omega}_M], \boldsymbol{\Omega}' = [\boldsymbol{\omega}_1', \boldsymbol{\omega}_2', \cdots, \boldsymbol{\omega}_M'] \tag{10}$$

As a result, we obtain a complex-valued spectral kernel mapping. The real part of the output is denoted as $\Re(\Phi(\boldsymbol{x})) = \cos(\boldsymbol{\Omega}^\top \boldsymbol{x}) + \cos(\boldsymbol{\Omega}'^\top \boldsymbol{x})$, and the imaginary part is denoted as $\Im(\Phi(\boldsymbol{x})) = \sin(\boldsymbol{\Omega}^\top \boldsymbol{x}) + \sin(\boldsymbol{\Omega}'^\top \boldsymbol{x})$.

**Complex-valued spectral kernel embedding module** In this module, we attempt to embed the complex-valued spectral kernel into each layer of neural networks to construct CosNet. Spectral kernel, based on the general Fourier analysis, provides a new explicit kernel mapping. These kernels can not only approximate most kernels under specific conditions by some fundamental theorems Cox and Miller [2017]; Yaglom [1987] but also provide an efficient way to combine neural networks with kernel methods to construct spectral networks. Most existing spectral kernel networks commonly embed the spectral kernel into neural networks by stacking the spectral kernel mapping in the hierarchical architecture of neural networks directly. However, the introduction of an imaginary part enables that networks with a simple stack of complex-valued mapping cannot be formulated as a spectral kernel (see the Supplementary Material for details).

To ensure the sub-network containing the first layer to arbitrary $l$-th layer ($l \geqslant 2$) can be integrally seen as a spectral kernel, and following the form of complex-valued parameters in CVNNs, we define the complex-valued weight matrix of this module as:

$$\boldsymbol{W} = \cos(\boldsymbol{A}) + i\sin(\boldsymbol{A}), \tag{11}$$

where $\boldsymbol{A}$ is a real-valued matrix.

In this module, the convolution operates with the complex weight matrix $\boldsymbol{W}$ is defined as:

$$\Psi(\boldsymbol{h}) = \boldsymbol{W}\boldsymbol{h} = \sqrt{\frac{1}{4M}}\Big[\cos(\boldsymbol{A})(\cos(\boldsymbol{\Omega}^\top \boldsymbol{x}) + \cos(\boldsymbol{\Omega}'^\top \boldsymbol{x}))$$
$$- \sin(\boldsymbol{A})(\sin(\boldsymbol{\Omega}^\top \boldsymbol{x}) + \sin(\boldsymbol{\Omega}'^\top \boldsymbol{x}))\Big]$$
$$+ i\sqrt{\frac{1}{4M}}\Big[\sin(\boldsymbol{A})(\cos(\boldsymbol{\Omega}^\top \boldsymbol{x}) + \cos(\boldsymbol{\Omega}'^\top \boldsymbol{x}))$$
$$+ \cos(\boldsymbol{A})(\sin(\boldsymbol{\Omega}^\top \boldsymbol{x}) + \sin(\boldsymbol{\Omega}'^\top \boldsymbol{x}))\Big]. \tag{12}$$

The real and imaginary parts of the convolution operation are represented in the matrix notation:

$$\begin{bmatrix} \Re(\Psi(\boldsymbol{h})) \\ \Im(\Psi(\boldsymbol{h})) \end{bmatrix} = \begin{bmatrix} \cos(\boldsymbol{A}) & -\sin(\boldsymbol{A}) \\ \sin(\boldsymbol{A}) & \cos(\boldsymbol{A}) \end{bmatrix} * \sqrt{\frac{1}{4M}} \begin{bmatrix} \cos(\boldsymbol{\Omega}^\top \boldsymbol{x}) + \cos(\boldsymbol{\Omega}'^\top \boldsymbol{x}) \\ \sin(\boldsymbol{\Omega}^\top \boldsymbol{x}) + \sin(\boldsymbol{\Omega}'^\top \boldsymbol{x}) \end{bmatrix}. \tag{13}$$

To inherit the outstanding representation capability from neural networks, in this module, we construct the spectral kernel networks by stacking $\Psi$:

$$CosNet(\boldsymbol{x}) = \Psi^{l-1}(\ldots \Psi^1(\boldsymbol{h})), \tag{14}$$

where $\Psi^l(2 \leq l)$ denotes the $l$-layer complex-valued spectral kernel mapping and

$$\boldsymbol{h} = \sqrt{\frac{1}{4M}}\Big[(\cos(\boldsymbol{\Omega}^\top \boldsymbol{x}) + \cos(\boldsymbol{\Omega'}^\top \boldsymbol{x})) \quad +i(\sin(\boldsymbol{\Omega}^\top \boldsymbol{x}) + \sin(\boldsymbol{\Omega'}^\top \boldsymbol{x}))\Big]. \tag{15}$$

### 3.4 Analysis of CosNet

CosNet, constructed by stacking the non-station complex-valued spectral kernel mapping, not only retains the property of non-stationary spectral kernels, which can effectively reveal the input-dependence characteristics and long-range relations but also can learn hierarchy within Reproducing Kernel Hilbert Space, yielding a cascade of non-linear features. Besides, CosNet takes the imaginary part of the complex-valued spectral kernel mapping into account, leading to a richer representation capability.

**Framework generality** In spectral kernels view, CosNet will be reduced to a stationarity spectral kernel when $\boldsymbol{\omega} = \boldsymbol{\omega'}$ in Equation (2). Besides, the real-valued spectral kernel mapping is the special case (*i.e.,* the imaginary part $\Im\Phi(\boldsymbol{x})$ equal to 0) of our complex-valued mapping. In the data view, CosNet can analyze the real-valued data, where the complex-valued representation can be found inherently. The first module of CosNet also can be considered as a complex-valued representation learning module, which transforms real-valued data into complex-valued features by optimizing the learnable frequency matrices $\boldsymbol{\Omega}$ and $\boldsymbol{\Omega'}$. CosNet also can analyze complex-valued data with the framework that only includes the second module.

**Parameters** In the complex-valued spectral kernel embedding module, we initialize the real and imaginary parts of weight matrices with the cosine and sine functions, respectively. Compared with CVNNs, which define the weight matrix as $\boldsymbol{W} = \boldsymbol{A} + i\boldsymbol{B}$, the number of parameters used in CosNet decreases because of our periodic initialization strategy using only $\boldsymbol{A}$. Compared to non-stationary spectral kernels, the number of parameters is reduced since there is no need for sampling two different frequency matrices, $\boldsymbol{\Omega}$ and $\boldsymbol{\Omega'}$.

**Theoretical results** We provide theoretical evidence of the generalization performance of CosNet, showing that CosNet has a lower generalization error bound compared to the real-valued spectral kernel networks of the same architecture. Concretely, we first bound the covering numbers of different layers in CosNet, followed by comparisons between covering numbers of real-valued spectral kernel networks and that of CosNet, which provide evidence of CosNet's improvements in generalization ability. Further, we derive the generalization bound of CosNet based on several theorems Bartlett *et al.* [2017]; Mohri *et al.* [2018].

**Theorem 1.** *Denote the covering number of set $S$ as $N_d(S, \epsilon)$. $\boldsymbol{X} \in R^{d^x \times n}$ is the input of $n$ samples and each sample is $d^x$-dimensioned. $\boldsymbol{X}^l \in R^{d^l \times n}$ is the input of layer $l$ $(l > 1)$ and $\boldsymbol{A}^l$ is the weight matrix of layer $l$ $(l \geq 2)$. The other notations remain the same as mentioned above. For different layers, their covering numbers satisfy that*

*1. In the first layer, $N_d(\boldsymbol{\Omega}^1 \boldsymbol{X}, \epsilon) \leq (4d^0 d^x)^k$, where $k \geq \frac{||\boldsymbol{\omega}_{ij}||_1^2}{\epsilon^2} \max_{i,j} ||\boldsymbol{x}_{ij}||^2$.*

*2. In layer $l$ $(l > 1)$, $N_d(\boldsymbol{A}^l \boldsymbol{X}^{l-1}, \epsilon) \leq (2d^l d^{l-1} + 1)^k$, where $k \geq \frac{||\boldsymbol{W}_{ij}||_1^2}{\epsilon^2} \frac{\pi}{2} d^l ||\boldsymbol{X}^{l-1}||_1^2$.*

*Proof.* The proof is relegated to the supplementary material of our paper due to space limitations. □

Covering numbers also serves as an indicator of models' representation ability, where the larger the covering number the greater the representation ability, but the more difficult it is to get the optimal solution. Note that when the weight matrices are the same, the bound of the covering number of each layer of a multilayer perception (MLP) is $(2d^l d^{l-1})^k$, where $k \geq \frac{||\boldsymbol{W}_{ij}||_1^2}{\epsilon^2} ||\boldsymbol{X}^{l-1}||_1^2$. And that of real-valued spectral network is $(4d^l d^{l-1})^k$, where $k \geq \frac{||\boldsymbol{W}_{ij}||_1^2}{\epsilon^2} ||\boldsymbol{X}^{l-1}||_1^2$, which is as twice large as that of MLP. It can be observed that real-value spectral networks improve their representation ability at the cost of much larger covering number bounds and poorer generalization performance. However,

CosNet combines the advantages of both MLP and real-valued spectral networks. Compared to MLP, CosNet's representation ability is further improved by bringing complex-valued representations into the spectral kernel networks, while only the covering number bound of the first layer increases when constant terms are neglected, which has stronger characterization ability and makes it easier to find the optimal solution. Its superiority is even more clear when compared to real-valued spectral kernels such as DSKN, every layer of CosNet has a smaller complexity, which leads to a significant difference when it comes to the complexity of the whole network.

**Theorem 2.** *Let $S = \{(\boldsymbol{x}_1, \boldsymbol{y}_1), (\boldsymbol{x}_2, \boldsymbol{y}_2), ..., (\boldsymbol{x}_n, \boldsymbol{y}_n)\}$ be a sample data of size $n$ from distribution $D$. Given the weight matrices defined before ($\boldsymbol{\Omega}^1, \boldsymbol{\Omega}^2, \boldsymbol{A}^1, \boldsymbol{A}^2, ..., \boldsymbol{A}^L$), and they satisfy that $||\boldsymbol{A}^l|| \leq c_l$, $||\boldsymbol{W}^l|| \leq b_l$, $||\boldsymbol{\Omega}^l|| \leq a_l$, $||\boldsymbol{X}||_1 \leq B$, $d^l \leq W$ and $T = (\sum_{l=1}^L (\frac{b_l}{c_l})^{2/3})^{3/2} \prod_{l=1}^L c_l$. And the loss function $\mathcal{L}(CosNet(\boldsymbol{x}), \boldsymbol{y}) \leq M$. Then with the probability of at least $1 - \delta$, the proposed network CosNet satisfies:*

$$
\begin{aligned}
&\underset{(\boldsymbol{x}, \boldsymbol{y}) \sim D}{E}[(\mathcal{L}(CosNet(\boldsymbol{x}), \boldsymbol{y})] \\
&\leq \frac{1}{n} \sum_{i=1}^n \mathcal{L}(CosNet(\boldsymbol{x}_i), \boldsymbol{y}_i) + \mathcal{O}(\frac{8M}{n^{3/2}} + M\sqrt{\frac{ln(1/\delta)}{n}} \\
&+ ln(n) \frac{\sqrt{ln(\boldsymbol{W})\boldsymbol{W}||\boldsymbol{X}^0||^2 T^2 + ln(\boldsymbol{W}) a_1^2 ||\boldsymbol{X}||^2}}{n}).
\end{aligned}
\tag{16}
$$

*Proof.* The proof is relegated to the supplementary material of our paper due to space limitations. □

## 4 Experiments

In this section, we first introduce the implementation details containing comparison methods and evaluation datasets. Then we conduct systematical experiments to demonstrate the superiority of the proposed CosNet, especially on the time series classification task.

**Datasets** To systematically evaluate the performance of our CosNet, we conduct comparison experiments on several typical time-series datasets, including 12 sub-datasets with default training and testing data splitting from the **UCR Archive** Dau *et al.* [2019] dataset for the classification task and 3 **UCI** Blake [1998] localization datasets for regression task. The overall statistics of the used datasets are shown in the Supplementary Material.

**Compared methods** We compare the proposed CosNet with several mainstream kernel methods and CVNNs, as follows: **SRFF** Zhang *et al.* [2017a]: Stacked Kernel Network, which stacks random Fourier features with stationary kernels; **DSKN** Xue *et al.* [2019]: Deep Spectral Kernel Network; **DCN** Trabelsi *et al.* [2018]: Deep Complex Network. We compare two variants with different commonly used activation functions, including $\mathbb{C}$ReLU (**DCN**[1]) and modReLU (**DCN**[2]); **ASKL** Li *et al.* [2020]: Automated spectral kernel learning.

**Implementation details** All the experiments are implemented with PyTorch Paszke *et al.* [2019] and conducted on a workstation with NVIDIA RTX 3090 GPU, AMD R7-5700X 3.40GHz 8-core CPU, and 32 GB memory. Each method is trained by ADAM Kingma and Ba [2014] using cross-entropy loss for the classification task and L2 loss for the regression task. The learning rate equals 0.01, and the weight matrix is initialized from a normal distribution $\mathcal{N}(0, 0.01)$. Each model contains five layers, including the input layer, the output layer, and three hidden layers. As exemplified by the time series classification task, the input is a time series (*i.e.* vector) with a scalar at each time point. The output is the implied feature mapping (*i.e.* vector), which is used to conduct the classification task. Concretely, the operation in the first layer is defined as $\Phi : \mathbb{R}^{d^x} \to \mathbb{C}^{d^x}$, where $d^x$ denotes the dimension of the data. Via $\Phi$ in the first layer, the data result in complex-valued representations, which are fed into the CSKE module starting from the second layer. The operation of $l^{th}$ layer is defined as $\Psi^l : \mathbb{C}^{d^l} \to \mathbb{C}^{d^{l+1}}$, where $d^l$ denotes the number of hidden complex-valued neuron. After the CSKE module, we obtain the implied complex-valued feathers. Moreover, these implied complex-valued features are condensed into vector form by the operation $\mathbb{C}^{d^L} \to \mathbb{R}^{2d^L}$, which concatenate real and imaginary parts, to conduct the classification task. Each experiment is repeated twenty times with different random seeds. Note that, the width of networks in each dataset depends

on the length of the time series, respectively. Therefore, the detailed settings of different models are shown in the Supplementary Material.

## 4.1 Experimental results

**Inherently complex-valued representation learning**     In practice, the observed data is always presented as real numbers, while the complex-valued representation can be found inherently in information processing. In CosNet, we propose the CSKG module, which generalizes the spectral kernel mapping in the real number domain to the complex number domain. To show the capability of our method to recover the complex-valued representation, we conduct a simulation experiment and compare CosNet with two typical strategies to deal with complex-valued mapping, i.e., Fourier transform (FT) and eliminating the complex part (DSKN). In the experiment, a complex number sequence with 100 points $\{z_i = u_i + v_i\}_{i=1}^{100}$, is randomly generated as the ground truth, and the corresponding real number is given as $x_i = |z_i|$. The first module of CosNet, i.e., CSKG, is used for recovering the given complex numbers $\{z_i\}_{i=1}^{100}$ from the real numbers $\{x_i\}_{i=1}^{100}$. The results are shown in Figure 2, from which we can find that CosNet with the complex-valued non-stationary spectral kernel mapping, compared with FT and DSKN, can recover the inherently complex-valued representation precisely. In contrast, the oscillations of the recovered sequences in both two domains by FT are intense while the DSKN fails to recover the phase information contained in the imaginary part.

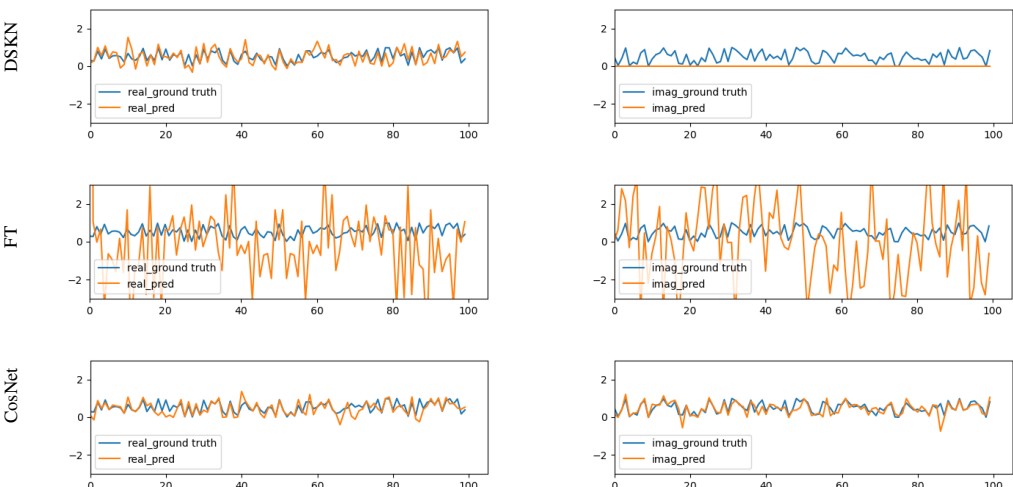

Figure 2: Comparison of complex-valued representation learning. The left and right denote the learning of real and imaginary parts, respectively.

**Time series classification**     To verify the effectiveness of our CosNet on the time-sequential data analysis, we compared the state-of-the-art spectral kernel networks and CVNNs on the time series classification task. The results are shown in Table 1. We can observe that our CosNet achieves state-of-the-art performance in all datasets. Specifically, CosNet outperforms other methods impressively and achieves 3% accuracy increment ($83.06 \rightarrow 85.46$) on Wine and 2.7% ($69.81 \rightarrow 71.73$) on FordB compared with the mainstream kernel methods and CVNNs. Furthermore, these results reveal that the performance of real-valued methods is limited to the data with the inherently complex-valued representation.

**Image classification and compression**     To further explore the representation capacity of our CosNet, we expand the application of CosNet to convolutional networks (see the Supplementary Material for details) for image classification and compression tasks using **Fashion-MNIST** Xiao *et al.* [2017] and **CIFAR10** Krizhevsky and Hinton [2009] datasets. For the classification task, accuracy is applied to the metric of performance. For the compression task, we first extract the implicit features through various models, and then we conduct the clustering task based on these extracted features. In this task, Normalized Mutual Information (NMI) and Rand Index (RI) are used as the assessment metrics. All the results are reported in Table 2. We can find out that our CosNet outperforms the baseline methods on both classification task and compression task. Notably, our CosNet achieves

Table 1: Classification accuracy (%) of each compared method on several time series datasets. The best results are highlighted in **bold**.

| Dataset | SRFF | DSKN | DCN$^1$ | DCN$^2$ | ASKL | CosNet |
|---|---|---|---|---|---|---|
| FordA | 81.46 | 82.24 | 81.87 | 79.90 | 72.66 | **82.42** |
| FordB | 68.99 | 69.81 | 69.68 | 50.17 | 64.20 | **71.73** |
| PhalangesOutlinesCorrect | 68.77 | 69.73 | 68.91 | 67.63 | 68.65 | **70.79** |
| Wine | 77.22 | 76.48 | 83.06 | 80.00 | 67.41 | **85.46** |
| ECG200 | 73.40 | 77.80 | 89.80 | 89.85 | 87.53 | **90.10** |
| ECG5000 | 91.98 | 91.14 | 93.11 | 93.50 | 92.75 | **93.70** |
| Herring | 57.73 | 56.64 | 65.23 | 58.13 | 59.52 | **65.39** |
| Ham | 51.52 | 48.81 | 71.10 | 67.76 | 68.52 | **71.29** |
| ProximalPhalanxOutlineAgeGroup | 79.49 | 79.51 | 81.80 | 81.59 | 80.00 | **82.71** |

Table 2: Classification and compression results on image datasets. The best results are highlighted in **bold**.

| | Fashion-MNIST | | | CIFAR10 | | |
|---|---|---|---|---|---|---|
| | DCN$^1$ | DCN$^2$ | CosNet | DCN$^1$ | DCN$^2$ | CosNet |
| Accuracy (%) | 87.02 | 84.44 | **88.33** | 64.32 | 52.39 | **66.51** |
| NMI (%) | 86.04 | 81.31 | **90.86** | 57.14 | 41.69 | **66.18** |
| RI (%) | 97.15 | 95.97 | **98.31** | 87.78 | 84.07 | **89.97** |

1.5% accuracy improvment (87.02% $\to$ 88.33%), 5.6% NMI improvement (86.04% $\to$ 90.86%), 1.19% RI improvement (97.15% $\to$ 98.13%) on FMNIST dataset, and 3.4% accuracy improvement (64.32% $\to$ 66.51, 15.82% NMI improvement (57.14% $\to$ 66.18%), 2.5% RI improvement (88.78% $\to$ 89.97%) on CIFAR-10 dataset. The results show that our CosNet has a greater representation capability than other complex-valued convolutional networks.

## 4.2 Ablation Study

**Complex-valued representation capability**    Complex numbers, containing amplitude and phase information, lead to a rich representation capability. However, the re-introducing of complex values brings extra parameters. To demonstrate that performance improvement comes from the powerful representation brought by complex values instead of added parameters, we conduct an ablation study to evaluate the methods with different parameters. As shown in Table 3, we compare the proposed CosNet with three variants of SRFF and DSKN, i.e., the variant in the original paper (normal), the variant with more neurons per layer (wider), and the variant with more layers (deeper).

From Table 3, we can observe that our CosNet commonly performs better with fewer parameters. Without loss of generality, increasing parameters indeed can lead to the improvement of performance on certain datasets, but there still remains a gap compared to our CosNet. Therefore, we can conclude that using real-valued networks in some fields, where complex numbers occur either naturally or by design, still has limitations.

**Initialization**    We propose to initialize the complex-valued weight matrix of the second module with the cosine and sine functions for the imaginary and real parts, respectively. This design ensures CosNet retains the property of non-stationary spectral kernels and takes the relative distance of data in the complex number domain without increasing the number of parameters. To explore the role of the designed initialization scheme, we compare the results of classification and regression tasks on CosNet with or without cosine and sine functions. The results reported in Table 4 show that our proposed initialization scheme performs better in all cases, which indicates that non-stationarity is necessary for analyzing the time-sequential data. Furthermore, the experimental results validate the effectiveness of our design in a complex-valued weight matrix on CosNet.

Table 3: Classification accuracy (%) and parameters with wider and deeper cases. The best results are highlighted in **bold**.

| Model | Setting | ECG200 | | ECG5000 | | Ham | |
|---|---|---|---|---|---|---|---|
| | | Parameters | Accuracy | Parameters | Accuracy | Parameters | Accuracy |
| SRFF | normal | 22.25K | 73.40 | **31.49K** | 91.98 | **251.12K** | 68.43 |
| | wider | 69.06K | 83.90 | 84.98K | 92.75 | 686.96K | 70.52 |
| | deeper | 42.35K | 65.85 | 56.91K | 92.55 | 459.22K | 60.10 |
| DSKN | normal | 44.42K | 77.80 | 62.80K | 91.14 | 502.11K | 69.76 |
| | wider | 137.99K | 80.65 | 169.62K | 92.42 | 1260.00K | 71.76 |
| | deeper | 137.99K | 80.65 | 169.62K | 92.42 | 1260.00K | 71.76 |
| CosNet | normal | **19.65K** | **90.10** | 40.75K | **93.70** | 375.14K | **74.27** |

Table 4: Classification accuracy (%)and regression MSE on the benchmark datasets. ($\uparrow$) indicates the larger the better, while ($\downarrow$) indicates the smaller the better. The best results are highlighted in **bold**.

| | Classification Accuracy ($\uparrow$) | | | Regression MSE ($\downarrow$) | | |
|---|---|---|---|---|---|---|
| | Earthquakes | DistalPhalanxTW | Strawberry | power | concreat | yacht |
| w/ cos, sin | **71.76** | **63.60** | **97.22** | **0.8229** | **1.3606** | **3.6270** |
| w/o cos, sin | 69.93 | 63.02 | 96.80 | 0.8795 | 1.3731 | 3.7932 |

## 5  Conclusion

In this paper, we propose a complex-valued spectral kernel network (CosNet) with two core modules, i.e., SKMG module and CSKE module. Specifically, as the first module of CosNet, the SKMG module is employed to recover the inherent complex-valued representation of the real-valued data. The CSKE module, designed by embedding the complex-valued spectral kernel mapping into neural networks with our initialization scheme, is used to effectively capture long-range or periodic relations of data. Our proposed CosNet, benefiting from the non-stationary property of kernels, can effectively encode the dynamic input-dependent characteristics and long-range correlations. The complex-valued mapping can improve the representation capacity of models without increasing the number of parameters. Furthermore, CosNet involves the transformation of the real-valued inputs in the optimization process to learn an expressive complex-valued representation. Moreover, some theoretical analyses of CosNet are also presented. Detailed experiments reveal that our proposed approach indeed leads to significant performance improvements over state-of-the-art relevant methods. Future work will focus on promoting the proposed CosNet in more applications.

**Limitation**   CosNet with the periodic function is prone to local minima. However, our CosNet tends to perform well in the time-sequential data analysis since it can not only capture the long-range relation in an input-dependent manner but also take the imaginary part into account. In future work, we will focus on promoting the proposed CosNet in the optimization method.

## Acknowledgments

This work was supported by the National Natural Science Foundation of China (Nos. 62076062 and 62306070) and the Social Development Science and Technology Project of Jiangsu Province (No. BE2022811). Furthermore, the work was also supported by the Big Data Computing Center of Southeast University.

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
