# Supplementary Materials of "CosNet: A Generalized Spectral Kernel Network"

**Yanfang Xue**[1,2], **Pengfei Fang**[1,2], **Jinyue Tian**[1,2], **Shipeng Zhu**[1,2], **Hui Xue**[1,2*]

[1]School of Computer Science and Engineering, Southeast University, Nanjing, 210096, China
[2]Key Laboratory of New Generation Artificial Intelligence Technology and Its Interdisciplinary Applications (Southeast University), Ministry of Education, China
{230218795, fangpengfei, 220222083, shipengzhu, hxue}@seu.edu.cn

In the supplementary material, we provide:

- Detailed proof of theoretical results (mentioned in the CosNet analysis part of the main paper).
- More explaination of the initialization scheme (mentioned in the complex-valued spectral kernel network part of the main paper)
- More details of the experiment and ablation studies (mentioned in the ablation study part of the main paper).

## 1 Proof of Theoretical Results

In this section, we collect the proof of the error bound omitted from the main paper. Before the proof, we introduce the necessary preliminary knowledge and notations.

**Foundation setting**    The proposed network with $L$ layers consists of two modules, i.e. SKMG module and CSKE module. The SKMG module includes 1 layer and CSKE model includes $L - 1$ layers. The input $\boldsymbol{X} \in \mathbb{R}^{d^x \times n}$ has $n$ samples and each sample $\boldsymbol{x}_i$ is $d^x$-dimension.

For the SKMG module, the weight matrix of the first linear transformation is denoted as $\boldsymbol{\Omega}^1 = \begin{bmatrix} \boldsymbol{\omega} \\ \boldsymbol{\omega}' \\ \boldsymbol{\omega} \\ \boldsymbol{\omega}' \end{bmatrix} \in \mathbb{R}^{4d^0 \times d^x}$, where $\boldsymbol{\omega}, \boldsymbol{\omega}' \in \mathbb{R}^{d^0 \times d^x}$. The second weight matrix is $\boldsymbol{\Omega}^2 =$

$\begin{bmatrix} \frac{1}{\sqrt{4d^0}}\boldsymbol{I}_{d^0} & \frac{1}{\sqrt{4d^0}}\boldsymbol{I}_{d^0} & 0 * \boldsymbol{J} & 0 * \boldsymbol{J} \\ 0 * \boldsymbol{J} & 0 * \boldsymbol{J} & \frac{1}{\sqrt{4d^0}}\boldsymbol{I}_{d^0} & \frac{1}{\sqrt{4d^0}}\boldsymbol{I}_{d^0} \end{bmatrix} \in \mathbb{R}^{2d^0 \times 4d^0}$, where $\boldsymbol{I}_{d^0} \in \mathbb{R}^{d^0 \times d^0}$ is the identity matrix and $\boldsymbol{J} \in \mathbb{R}^{d^0 \times d^0}$ is a matrix of ones. The output of this module is denoted as $\boldsymbol{X}^0 \in \mathbb{R}^{d^0 \times n}$.

For the CSKE module, the input is denoted as $\boldsymbol{X}^{l-1}$ with $d^{l-1}$-dimension and the output is denoted as $\boldsymbol{X}^l$ in the $l^{th}$ layer. Note that: $\boldsymbol{X}^1 = \boldsymbol{X}^0, d^1 = d^0$, which means the output of the SKMG module is used as the input of CSKE module. The weight matrix of $l^{th}$ ($2 \leq l \leq L$) layer is denoted as

$$\mathbf{W}^l = \begin{bmatrix} \cos(\boldsymbol{A}^l) & -\sin(\boldsymbol{A}^l) \\ \sin(\boldsymbol{A}^l) & \cos(\boldsymbol{A}^l) \end{bmatrix} \in \mathbb{R}^{2d^l \times 2d^{l-1}},$$

where $\boldsymbol{A}^l \in \mathbb{R}^{d^l \times d^{l-1}}$.

**Definition 1.** *($\epsilon$-net) $A$'s subset $\widetilde{A}$ is an $\epsilon$-net of $A$ under the metric $d$ if for any $a \in A$ there exists $\widetilde{a} \in \widetilde{A}$ that $d(a, \widetilde{a}) \leq \epsilon$.*

---

*Corresponding author

37th Conference on Neural Information Processing Systems (NeurIPS 2023).

**Definition 2.** *(covering number) The covering number $N_d(A, \epsilon)$ is the size of the smallest $\epsilon$-net of $A$.*

**Lemma 1.** *(Maury's sparsification lemma Pisier). Fix Hilbert space $\mathcal{H}$ with norm $|| \cdot ||$. Let $\boldsymbol{U} \in \mathcal{H}$ be given with representation $\boldsymbol{U} = \sum_{i=1}^{d} \alpha_i \boldsymbol{V}_i$, where $\boldsymbol{V}_i \in \mathcal{H}$ and $\boldsymbol{\alpha} = [\alpha_1, \alpha_2, ..., \alpha_d] \in \mathbb{R}_{\geq 0}^d \backslash \{0\}$. Then for any positive integer $k$, there exists a choice of non-negative integers $(k_1, ..., k_d)$, $\sum_{i=1}^{d} k_i = k$, such that*

$$||\boldsymbol{U} - \frac{||\boldsymbol{\alpha}||_1}{k} \sum_{i=1}^{d} k_i \boldsymbol{V}_i||^2 \leq \frac{||\boldsymbol{\alpha}||_1}{k} \sum_{i=1}^{d} \alpha_i ||\boldsymbol{V}_i||^2 \leq \frac{||\boldsymbol{\alpha}||_1^2}{k} \max_i ||\boldsymbol{V}_i||^2.$$

*Proof.* Set $\beta = ||\boldsymbol{\alpha}||_1$, and let $(\boldsymbol{B}_1, \boldsymbol{B}_2, ..., \boldsymbol{B}_k)$ denote $k$ iid random variables where $P(\boldsymbol{B}_1 = \beta \boldsymbol{V}_i) = \frac{\alpha_i}{\beta}$. Define $\boldsymbol{B} = \frac{1}{k} \sum_{i=1}^{k} \boldsymbol{B}_i$, whereby

$$\mathbb{E}[\boldsymbol{B}] = \mathbb{E}[\frac{1}{k} \sum_{i=1}^{k} \boldsymbol{B}_i] = \frac{1}{k} \mathbb{E}[k \boldsymbol{B}_1] = \mathbb{E}[\boldsymbol{B}_1] = \sum_{i=1}^{d} (\beta \boldsymbol{V}_i) \frac{\alpha_i}{\beta} = \sum_{i=1}^{d} \alpha_i \boldsymbol{V}_i = \boldsymbol{U}.$$

Consequently,

$$\mathbb{E}[||\boldsymbol{U} - \boldsymbol{B}||^2] = \frac{1}{k^2} \mathbb{E}[|| \sum_{i=1}^{d} (\boldsymbol{U} - \boldsymbol{B}_i)||^2] = \frac{1}{k^2} \mathbb{E}[\sum_i ||\boldsymbol{U} - \boldsymbol{B}_i||^2 + \sum_{i \neq j} < \boldsymbol{U} - \boldsymbol{B}_i, \boldsymbol{U} - \boldsymbol{B}_j >]$$

$$= \frac{1}{k} \mathbb{E}[||\boldsymbol{U} - \boldsymbol{B}_1||^2] = \frac{1}{k} (\mathbb{E}[||\boldsymbol{B}_1||^2] - ||\boldsymbol{U}||^2) \leq \frac{1}{k} \mathbb{E}[||\boldsymbol{B}_1||^2]$$

$$= \frac{1}{k} \sum_{i=1}^{d} \frac{\alpha_i}{\beta} ||\beta \boldsymbol{V}_i||^2 = \frac{\beta}{k} \sum_{i=1}^{d} \alpha_i ||\boldsymbol{V}_i||^2$$

$$\leq \frac{\beta^2}{k} \max_i ||\boldsymbol{V}_i||^2.$$

By the probabilistic method, there exists intergers $(j_1, .., j_k) \in \{1, .., d\}^k$ and an assignment $\widetilde{\boldsymbol{B}}_i = \beta \boldsymbol{V}_{j_i}$ and $\widetilde{\boldsymbol{B}} = \frac{1}{k} \sum_{i=1}^{k} \widetilde{\boldsymbol{B}}_i$ such that

$$||\boldsymbol{U} - \widetilde{\boldsymbol{B}}||^2 \leq E[||\boldsymbol{U} - \boldsymbol{B}||^2].$$

The result now follows by defining integers $(k_1, ..., k_d)$ according to $k_i = \sum_{l=1}^{k} \mathbb{I}_{[j_l = i]}$, where $\mathbb{I}(\cdot)$ denotes the indicator function. $\square$

**Corollary 1.** *Fix Hilbert space $\mathcal{H}$ with norm $|| \cdot ||$. Let $\boldsymbol{U} \in \mathcal{H}$ be given with representation $\boldsymbol{U} = \sum_{i=1}^{d} \alpha_i \boldsymbol{V}_i$, where $\boldsymbol{V}_i \in \mathcal{H}$ and $\boldsymbol{\alpha} = [\alpha_1, \alpha_2, ..., \alpha_d] \in \mathbb{R}_{\geq 0}^d \backslash \{0\}$. Then for any positive integer $k$ and for any $m \geq \alpha_i (\forall i)$, there exists a choice of non-negative integers $(k_1, ..., k_d)$, $\sum_{i=1}^{d} k_i = k$ such that*

$$||\boldsymbol{U} - \frac{m}{k} \sum_{i=1}^{d} k_i \boldsymbol{V}_i||^2 \leq \frac{m}{k} \sum_{i=1}^{d} \alpha_i ||\boldsymbol{V}_i||^2 \leq \frac{m^2}{k} \max_i ||\boldsymbol{V}_i||^2.$$

*Proof.* Following the proof of Lemma 1 with $\beta = m$, the result is trivial. $\square$

**Corollary 2.** *For set $A = \{\sum_{i=1}^{d} \alpha_i \boldsymbol{V}_i | \boldsymbol{\alpha} \in \mathbb{R}_{\geq 0}^d \backslash \{0\}, ||\boldsymbol{\alpha}||_1 \leq \widetilde{\alpha}\} \in conv(\boldsymbol{V}_1, \boldsymbol{V}_2, ..., \boldsymbol{V}_d)$, its covering number satisfies $N_d(A, \epsilon) \leq d^k$ and $\ln(N_d(A, \epsilon)) \leq k \ln(d)$, where $k$ is an integer and $k \geq \frac{\widetilde{\alpha}^2}{\epsilon^2} \max_i ||\boldsymbol{V}_i||^2$.*

*Proof.* For any $\boldsymbol{U} \in A$, there exist $\boldsymbol{\alpha}$ satisfying $\boldsymbol{U} = \sum_{i=1}^{d} \alpha_i \boldsymbol{V}_i$. Since $||\boldsymbol{\alpha}||_1 \leq \widetilde{\alpha}$, by Corollary 1, for a fixed positive integer $k$ satisfying $k \geq \frac{\widetilde{\alpha}^2}{\epsilon^2} max_i ||\boldsymbol{V}_i||^2$, there exists a choice of non-negative integers $(k_1, ..., k_d)$, $\sum_{i=1}^{d} k_i = k$, such that

$$||\boldsymbol{U} - \frac{\widetilde{\alpha}}{k} \sum_{i=1}^{d} k_i \boldsymbol{V}_i||^2 \leq \frac{\widetilde{\alpha}}{k} \sum_{i=1}^{d} \alpha_i ||\boldsymbol{V}_i||^2 \leq \frac{\widetilde{\alpha}_1^2}{k} max_i ||\boldsymbol{V}_i||^2 \leq \epsilon^2.$$

By Definition 1, $\{\frac{\widetilde{\alpha}}{k} \sum_{i=1}^{d} k_i \boldsymbol{V}_i | k_i (i = 1, .., d), \sum_{i=1}^{d} k_i = k\}$ is a $\epsilon$-net of $A$. With $k_i$ being non-negative integers and $\sum_{i=1}^{d} k_i = k$, the cardinality of this set satisfies

$$|\{\frac{\widetilde{\alpha}}{k} \sum_{i=1}^{d} k_i \boldsymbol{V}_i\}| \leq d^k.$$

By Definition 2, $N_d(A, \epsilon) \leq d^k$ and $\ln(N_d(A), \epsilon)) \leq k \ln(d)$, where $k$ is an integer and $k \geq \frac{\widetilde{\alpha}^2}{\epsilon^2} max_i ||\boldsymbol{V}_i||^2$.

$\square$

**Lemma 2.** *For the non-empty set $A \subset \mathbb{R}^n$, define a projection* $f(\boldsymbol{a}) = \begin{bmatrix} \sigma(a_1) \\ \sigma(a_2) \\ \vdots \\ \sigma(a_n) \end{bmatrix}$, *where $\boldsymbol{a} \in A$,*

$\boldsymbol{a} = \begin{bmatrix} a_1 \\ a_2 \\ \vdots \\ a_n \end{bmatrix}$, *$\sigma$ is a function. And the distance metric of this space is defined by p-norm. If $\sigma$ is*

*l-Lipschitz and $\widetilde{A}$ is a $\epsilon$-net of $A$, then $f(\widetilde{A})$ is a $\epsilon$-net of $f(A)$.*

*Proof.* Denote the distance metric in the space as $d(\boldsymbol{a}, \boldsymbol{a}') = ||\boldsymbol{a} - \boldsymbol{a}'||_p$.

Since $\widetilde{A}$ is a $\epsilon$-net of $A$, then for any $\boldsymbol{a} \in A$, there exists $\widetilde{\boldsymbol{a}} \in \widetilde{A}$ satisfying that $d(\boldsymbol{a}, \widetilde{\boldsymbol{a}}) \leq \epsilon$.

For any $\boldsymbol{f} \in f(A)$, there exists $\boldsymbol{a} \in A$ so that $f(\boldsymbol{a}) = \boldsymbol{f}$ and corresponding $\widetilde{\boldsymbol{a}}$ that $\widetilde{\boldsymbol{f}} = f(\widetilde{\boldsymbol{a}}) \in f(\widetilde{A})$ and $d(\boldsymbol{a}, \widetilde{\boldsymbol{a}}) \leq \epsilon$. Then,

$$d(\boldsymbol{f}, \widetilde{\boldsymbol{f}}) = d(f(\boldsymbol{a}), f(\widetilde{\boldsymbol{a}})) = (\sum_{i=1}^{n} (\sigma(\boldsymbol{a}_i) - \sigma(\widetilde{\boldsymbol{a}}_i))^p)^{1/p} = ld(\boldsymbol{a}, \widetilde{\boldsymbol{a}}) \leq l\epsilon \triangleq \epsilon'.$$

$\square$

**Corollary 3.** *Following the definitions in Lemma 2, if there exists a covering number for set $A$, then $N_d(f(A), \epsilon') \leq N_d(A, \hat{\epsilon})$.*

*Proof.* Assume $\widetilde{A}$ is an $\epsilon$-net of $A$.

By Lemma 2, $f(\widetilde{A})$ is an $\epsilon$-net of $\widetilde{A}$.

By Definition 2, since the cardinality: $|f(\widetilde{A})| \leq |\widetilde{A}|$, so $N_d(f(A), \epsilon') \leq N_d(A, \epsilon)$.

$\square$

**Theorem 1.** *Denote the covering number of set $S$ as $N_d(S, \epsilon)$. $\boldsymbol{X} \in \mathbb{R}^{d^x \times n}$ is the input of $n$ samples and each sample is $d^x$-dimensioned. $\boldsymbol{X}^l \in \mathbb{R}^{d^l \times n}$ is the input of layer $l$ ($l > 1$) and $\boldsymbol{W}^l$ is the weight matrix of layer $l$ ($l \geq 2$). The other notations remain the same as mentioned above. For different layers, their covering numbers satisfy that*

1. *In the first layer (i.e., the SKMG module), $N_d(\boldsymbol{\Omega}^1 \boldsymbol{X}, \epsilon) \leq (4d^0 d^x)^k$, where $k \geq \frac{||\boldsymbol{\Omega}_{ij}||_1^2}{\epsilon^2} max_{i,j} ||\boldsymbol{x}_{ij}||^2$.*

2. *In layer $l$ ($l > 1$) (i.e., the CSKE module), $N_d(\boldsymbol{W}^l \boldsymbol{X}^{l-1}, \epsilon) \leq (2d^l d^{l-1} + 1)^k$, where $k \geq \frac{||\boldsymbol{A}_{ij}||_1^2}{\epsilon^2} \frac{\pi}{2} d^l ||\boldsymbol{X}^{l-1}||_1^2$.*

*Proof.* Part 1

We denote $\boldsymbol{P}_{kq} \in \mathbb{R}^{d^0 \times d^x}$ as a matrix where only the element of row $k$ column $q$ is 1, other element is 0. Trivially, any $\boldsymbol{\Omega}^1 \in \mathbb{R}^{4d^0 \times d^x}$ can be written as:

$$\boldsymbol{\Omega}^1 = \boldsymbol{\alpha} \sum_{1 \leq k \leq d^0, 1 \leq q \leq d^x} \pm \begin{bmatrix} \boldsymbol{P}_{kq} \\ 0 * \boldsymbol{J} \\ \boldsymbol{P}_{kq} \\ 0 * \boldsymbol{J} \end{bmatrix} \pm \begin{bmatrix} 0 * \boldsymbol{J} \\ \boldsymbol{P}_{kq} \\ 0 * \boldsymbol{J} \\ \boldsymbol{P}_{kq} \end{bmatrix}.$$

For simplicity, denote the $4d^0 d^x$ components shown above as $\{\boldsymbol{V}_i\}_{i=1}^{4d^0 d^x}$, then $\boldsymbol{\Omega}^1 = \sum_{i=1}^{4d^0 d^x} \alpha_i \boldsymbol{V}_i$, where $\boldsymbol{\alpha} \in \mathbb{R}_{\geq 0}^d \backslash \{0\}$. And $\boldsymbol{\Omega}^1 \boldsymbol{X} = \sum_{i=1}^{4d^0 d^x} \alpha_i \boldsymbol{V}_i \boldsymbol{X} = \sum_{i=1}^{4d^0 d^x} \alpha_i \boldsymbol{V}_i'$, where $\boldsymbol{V}_i' = \boldsymbol{V}_i \boldsymbol{X}$.

By Corollary 2, $N_d(\boldsymbol{\Omega} \boldsymbol{X}, \epsilon) \leq (4d^0 d^x)^k$, where $k \geq \frac{||\boldsymbol{\Omega}_{ij}||_1^2}{\epsilon^2} \max_{i,j} ||x_{ij}||^2$.

Part 2

To analyze the covering number of $\boldsymbol{W}^l \boldsymbol{X}^{l-1}$, first begin with the covering number of $\boldsymbol{B}^l \boldsymbol{X}^{l-1}$, and $\boldsymbol{B}^l$ is defined as

$$\boldsymbol{B}^l = \begin{bmatrix} \boldsymbol{A}^l & \boldsymbol{A}^l + \frac{\pi}{2} * \boldsymbol{J} \\ \boldsymbol{A}^l - \frac{\pi}{2} * \boldsymbol{J} & \boldsymbol{A}^l \end{bmatrix}.$$

We rewrite $\boldsymbol{B}^l$ as

$$\boldsymbol{B}^l = \boldsymbol{\alpha} \left( \sum_{1 \leq k \leq d^l, 1 \leq q \leq d^{l-1}} \pm \begin{bmatrix} \boldsymbol{P}_{kq} & \boldsymbol{P}_{kq} \\ \boldsymbol{P}_{kq} & \boldsymbol{P}_{kq} \end{bmatrix} \right) + 1 \cdot \begin{bmatrix} 0 * \boldsymbol{J} & \frac{\pi}{2} * \boldsymbol{J} \\ -\frac{\pi}{2} * \boldsymbol{J} & 0 * \boldsymbol{J} \end{bmatrix},$$

where $\boldsymbol{P}_{kq}$ is in $\mathbb{R}^{d^l \times d^{l-1}}$.

For simplicity, denote the $2d^l d^{l-1} + 1$ components shown above as $\{\boldsymbol{V}_i\}_{i=1}^{2d^l d^{l-l}+1}$, then $\boldsymbol{B}^l = \sum_{i=1}^{2d^l d^{l-1}+1} \alpha_l \boldsymbol{V}_i$, where $\boldsymbol{\alpha} \in \mathbb{R}_{\geq 0}^d \backslash \{0\}$. And $\boldsymbol{B}^1 \boldsymbol{X}^l = \sum_{i=1}^{2d^l d^{l-1}+1} \alpha_i \boldsymbol{V}_i \boldsymbol{X}^{l-1} = \sum_{i=1}^{2d^l d^{l-1}+1} \alpha_i \boldsymbol{V}_i'$, where $\boldsymbol{V}_i' = \boldsymbol{V}_i \boldsymbol{X}^l$.

If $\boldsymbol{V}_i$ is in the form of $\begin{bmatrix} \boldsymbol{P}_{kq} & \boldsymbol{P}_{kq} \\ \boldsymbol{P}_{kq} & \boldsymbol{P}_{kq} \end{bmatrix}$, then

$$||\boldsymbol{V}_i'||_1 \leq max_{1 \leq k \leq d^{l-1}} 2 \sum_{j=1}^{n} |\boldsymbol{X}_{kj}^{l-1} + \boldsymbol{X}_{k+d^{l-1},j}^{l-1}|.$$

If $\boldsymbol{V}_i$ is in the form of $\begin{bmatrix} 0 * \boldsymbol{J} & \frac{\pi}{2} * \boldsymbol{J} \\ -\frac{\pi}{2} * \boldsymbol{J} & 0 * \boldsymbol{J} \end{bmatrix}$, then

$$||\boldsymbol{V}_i'||_1 \leq \frac{\pi}{2} d^l \sum_{j=1}^{n} [| \sum_{k=1}^{d^{l-1}} \boldsymbol{X}_{kj}^{l-1}| + | \sum_{k=d^{i-1}+1}^{2d^{i-1}} \boldsymbol{X}_{kj}^{l-1}|] \leq \frac{\pi}{2} d^l ||\boldsymbol{X}^{l-1}||_1.$$

In summary, since $\frac{\pi}{2} d^l ||\boldsymbol{X}^{l-1}||_1 > max_{1 \leq k \leq d^{l-1}} 2 \sum_{j=1}^{n} |\boldsymbol{X}_{kj}^{l-1} + \boldsymbol{X}_{k+d^{l-1},j}^{l-1}|$, by Corollary 2, $N_d(\boldsymbol{B}^l \boldsymbol{X}^{l-1}, \epsilon) \leq (2d^l d^{l-1} + 1)^k$, where $k \geq \frac{||\boldsymbol{A}_{ij}||_1^2}{\epsilon^2} \frac{\pi}{2} d^l ||\boldsymbol{X}^{l-1}||_1^2$.

Since $\boldsymbol{W}^l = \cos(\boldsymbol{B}^l)$, by Corollary 3, $N_d(\boldsymbol{B}^l \boldsymbol{X}^{l-1}, \epsilon) \leq (2d^l d^{l-1} + 1)^k$, where $k \geq \frac{||\boldsymbol{A}_{ij}||_1^2}{\epsilon^2} \frac{\pi}{2} d^l ||\boldsymbol{X}^{l-1}||_1^2$. $\square$

Then we introduce two important lemmas about Rademacher complexity and empirical Rademacher complexity respectively.

**Lemma 3.** *Mohri* et al. *[2018] Let $F_{|S}$ be a real-valued function class taking values in $[0, 1]$ given the dataset $S$, and assume that $0 \in F_{|S}$. Then the Rademacher complexity given the dataset $S$ satisfies that*

$$R(F_{|S}) \leq \inf_{\alpha > 0}(\frac{4\alpha}{\sqrt{n}} + \frac{12}{n} \int_{\alpha}^{\sqrt{n}} \sqrt{lnN_d(F_{|S}, \epsilon, ||\cdot||_2)}d\epsilon).$$

**Lemma 4.** *Mohri* et al. *[2018] Let $\mathcal{L} : X \times Y \to \mathbb{R}$ be an $L_p$ loss function bounded by $M > 0$, $\mathcal{F}$ be the hypothesis set, family $\mathcal{G} = \{(\boldsymbol{x}, \boldsymbol{y}) \to \mathcal{L}(F(\boldsymbol{x}), \boldsymbol{y}) : F \in \mathcal{F}\}$, then for any $\delta$, with probability at least $1 - \delta$, the following inequality holds:*

$$E_{(\boldsymbol{x},\boldsymbol{y}) \sim \mathcal{D}}[(\mathcal{L}(F(\boldsymbol{x}), \boldsymbol{y})] \leq \frac{1}{n} \sum_{i=1}^{n} l(F(\boldsymbol{x}_i), \boldsymbol{y}_i) + 2\hat{R}_S(\mathcal{G}) + 3M\sqrt{\frac{ln(\frac{2}{\delta})}{2n}}.$$

*where $\hat{R}_S(\mathcal{G})$ is the empirical Rademacher complexity of the family $\mathcal{G}$ given the dataset $S$.*

**Lemma 5.** *Let $(\epsilon_1, ..., \epsilon_L)$ be given, along with operator norm bounds $(c_1, .., c_L)$. Suppose the matrix $\boldsymbol{\Theta} = (\boldsymbol{\Omega}^1, \boldsymbol{\Omega}^2, \boldsymbol{W}^2, .., \boldsymbol{W}^L)$ lie within $B^1 \times B^2 \times ... \times B^{L+1}$ where $B^l$ are arbitrary classes with the property that each $\boldsymbol{W}^l \in B^l$ has $sup_{||\boldsymbol{x}|| \leq 1}||\boldsymbol{W}^l\boldsymbol{x}|| = c_l$. Lastly, let data $\boldsymbol{X}$ be given with $||\boldsymbol{X}||_1 \leq B$.Then, letting $\tau = \sum_{j \leq L} \epsilon_j \prod_{l=j+1}^{L} c_l$, the neural network hypothesis space $H_{\boldsymbol{X}} = \{F_{\boldsymbol{\Theta}}(\boldsymbol{X})|\boldsymbol{\Theta} \in B^1 \times B^2 \times ... \times B^{L+1}\}$ has covering number bound*

$$N_d(H_{\boldsymbol{X}}, \tau, |\cdot|_L) \leq \prod_{l=1}^{L} \sup_{(\boldsymbol{\Omega}^1.\boldsymbol{\Omega}^2, \boldsymbol{W}^2, .., \boldsymbol{W}^L), \forall j < i} N_d(\{W^l F_{(\boldsymbol{\Omega}^1, \boldsymbol{\Omega}^2, \boldsymbol{W}^2, ..., \boldsymbol{W}^L)}(\boldsymbol{X})\}, \epsilon_l, ||\cdot||_{l+1}).$$

*Proof.* It is proved by mathematical induction. Inductively construct covers $\mathcal{C}_l$ of $\boldsymbol{\Omega}^1\boldsymbol{X}, ..., \boldsymbol{W}^L\boldsymbol{X}^{L-1}$.

- When $l = 1$, since $\boldsymbol{\Omega}^2$ is fixed once the output dimension is chosen, it is trivial that the lemma holds.

- When $l = 2$

  Denote $\mathcal{C}_1$ as an $\epsilon$-net of $\boldsymbol{\Omega}^1\boldsymbol{X}$, then

  $$|\mathcal{C}_1| \leq N_d(\{\boldsymbol{\Omega}^1\boldsymbol{X} : \boldsymbol{\Omega}^1 \in B^1\}, \epsilon_1, ||\cdot||_2) \triangleq N_1.$$

  For a fixed $C \in \mathcal{C}_1$, there exists an $\epsilon$-net $G(C)$ that

  $$|G(C)| \leq N_d(\{\boldsymbol{\Omega}^2(\sigma(\boldsymbol{\Omega}^1\boldsymbol{X})) : \boldsymbol{\Omega}^2 \in B^2\}, \epsilon_2, ||\cdot||_2) \triangleq N_2.$$

  Set $\mathcal{C}_2 = \cup_{C \in \mathcal{C}_1} G(C)$, then $\mathcal{C}_2$ is an $\epsilon$-net of $\{\boldsymbol{\Omega}^2(\sigma(\boldsymbol{\Omega}^1\boldsymbol{X})) : \boldsymbol{\Omega}^2 \in B^2\}$.

  Then, $|\mathcal{C}_2| \leq N_1 N_2$.

  And for any $\boldsymbol{X}^0 \in \boldsymbol{\Omega}^2(\sigma(\boldsymbol{\Omega}^1\boldsymbol{X}))$, there exists $\hat{\boldsymbol{X}}^0 \in \mathcal{C}_2$ that:

  $$|\boldsymbol{X}^0 - \hat{\boldsymbol{X}}^0| = |\boldsymbol{\Omega}^2(\sigma(\boldsymbol{\Omega}^1(\boldsymbol{X}))) - \boldsymbol{\Omega}^2(\sigma(\widetilde{\boldsymbol{\Omega}^1(\boldsymbol{X}}))))|$$

  $$\leq |\boldsymbol{\Omega}^2(\sigma(\boldsymbol{\Omega}^1(\boldsymbol{X}))) - \boldsymbol{\Omega}^2(\sigma(\widetilde{\boldsymbol{\Omega}^1(\boldsymbol{X}})))| + |\boldsymbol{\Omega}^2(\sigma(\widetilde{\boldsymbol{\Omega}^1(\boldsymbol{X}}))) - \boldsymbol{\Omega}^2(\sigma(\widetilde{\boldsymbol{\Omega}^1(\boldsymbol{X}}))))|$$

  $$\leq |\boldsymbol{\Omega}^2||\sigma(\boldsymbol{\Omega}^1(\boldsymbol{X})) - \sigma(\widetilde{\boldsymbol{\Omega}^1(\boldsymbol{X}}))| + \epsilon_2$$

  $$\leq c_2\epsilon_1 + \epsilon_2.$$

  Note that $\hat{\boldsymbol{X}}^0 = \boldsymbol{\Omega}^2(\sigma(\widetilde{\boldsymbol{\Omega}^1(\boldsymbol{X}})))$ is in an $\epsilon$-net of $\boldsymbol{\Omega}^2(\sigma(\widetilde{\boldsymbol{\Omega}^1(\boldsymbol{X}})))$, and $\sigma(\widetilde{\boldsymbol{\Omega}^1(\boldsymbol{X}}))$ is in an $\epsilon$-net of $\sigma(\boldsymbol{\Omega}^1(\boldsymbol{X}))$.

  The lemma holds under this condition.

- Assume the lemma holds when $1 \leq l < L$.

- When $l = L$, use the same notation as above, set $\mathcal{C}_{L+1} = \cup_{C \in \mathcal{C}_L} G(C)$, then $|\mathcal{C}_{L+1}| \leq \prod_{l=1}^{L} N_l$.

And for any $\boldsymbol{X}^L$, there exists $\hat{\boldsymbol{X}}^L \in \mathcal{C}_{L+1}$ that:

$$
\begin{aligned}
|\boldsymbol{X}^L - \hat{\boldsymbol{X}}^L| &= |\boldsymbol{W}^L \boldsymbol{X}^{L-1} - \widetilde{\boldsymbol{W}^L \boldsymbol{X}^{L+1}}| \\
&\leq |\boldsymbol{W}^L \boldsymbol{X}^{L-1} - \boldsymbol{W}^L \widetilde{\boldsymbol{X}^{L-1}}| + |\boldsymbol{W}^L \widetilde{\boldsymbol{X}^{L-1}} - \widetilde{\boldsymbol{W}^L \boldsymbol{X}^{L-1}}| \\
&\leq |\boldsymbol{W}^L||\boldsymbol{X}^{L-1} - \widetilde{\boldsymbol{X}^{L-1}}| + \epsilon_{L+1} \\
&\leq c_L (\sum_{j \leq i} \epsilon_j \prod_{l=j+1}^{L-1} c_l) + \epsilon_L \\
&= \sum_{j \leq L} \epsilon_j \prod_{l=j+1}^{L} c_l.
\end{aligned}
$$

Note that $\hat{\boldsymbol{X}}^L = \widetilde{\boldsymbol{W}^L \boldsymbol{X}^{L-1}}$ is in an $\epsilon$-net of $\boldsymbol{W}^L \widetilde{\boldsymbol{X}^{L-1}}$, and $\widetilde{\boldsymbol{X}^{L-1}}$ is in an $\epsilon$-net of $\boldsymbol{X}^{L-1}$.

By induction, the lemma holds. $\qquad\square$

**Theorem 2.** *Let $S = \{(\boldsymbol{x}_1, \boldsymbol{y}_1), (\boldsymbol{x}_2, \boldsymbol{y}_2), ..., (\boldsymbol{x}_n, \boldsymbol{y}_n)\}$ be a sample data of size $n$ from distribution $\mathcal{D}$. Given the weight matrices defined before $(\boldsymbol{\Omega}^1, \boldsymbol{\Omega}^2, \boldsymbol{W}^2, ..., \boldsymbol{W}^L)$, and they satisfy that $||\boldsymbol{W}^l|| \leq c_l (l > 2)$, $||\boldsymbol{\Omega}^1|| \leq a_1$, $||\boldsymbol{\Omega}^2|| \leq c_1$, $||\boldsymbol{A}^l|| \leq b_l$, $||\boldsymbol{X}||_1 \leq B$, $d^l \leq W$ and $T = (\sum_{l=2}^{L}(\frac{b_l}{c_l})^{2/3})^{3/2} \prod_{l=1}^{L} c_l$. And the loss function $\mathcal{L}(F(\boldsymbol{x}), \boldsymbol{y}) \leq M$. Then, with the probability at least $1 - \delta$, the proposed network $F$ satisfy:*

$$
\begin{aligned}
\mathbb{E}_{(\boldsymbol{x}, \boldsymbol{y}) \sim \mathcal{D}}[(\mathcal{L}(F_{\boldsymbol{\Theta}}(\boldsymbol{x}), \boldsymbol{y})] &\leq \frac{1}{n} \sum_{i=1}^{n} \mathcal{L}(F_{\boldsymbol{\Theta}}(\boldsymbol{x}_i), \boldsymbol{y}_i) \\
&+ \mathcal{O}(\frac{8M}{n^{3/2}} + M\sqrt{\frac{ln(1/\delta)}{n}} + ln(n) \frac{\sqrt{ln(W)W||\boldsymbol{X}^0||^2 T^2 + ln(W)a_1^2||\boldsymbol{X}||^2}}{n}).
\end{aligned}
$$

*Proof.* Follow the notaion above, by Lemma 5 and Theorem 1, the covering number of the whole network has:

$$
\begin{aligned}
ln(N_d(H_X, \tau, |\cdot|_{L+1})) &\leq \sum_{l=1}^{L} \sup_{(\boldsymbol{\Omega}^1.\boldsymbol{\Omega}^2, \boldsymbol{W}^2, .., \boldsymbol{W}^j), \forall j < l} ln(N_d(\{\boldsymbol{W}^l F_{\boldsymbol{\Omega}^1, \boldsymbol{\Omega}^2, \boldsymbol{W}^2, ..., \boldsymbol{W}^{l-1}}(\boldsymbol{X})\}, \epsilon_l, ||\cdot||_{l+1})). \\
&\leq \sum_{l=1}^{L} \sup_{(\boldsymbol{\Omega}^1.\boldsymbol{\Omega}^2, \boldsymbol{W}^2, .., \boldsymbol{W}^L), \forall j < l} ln(2d^l d^{l-1} + 1)k^l + ln(4d^0 d^x)k^0,
\end{aligned}
$$

where, $k^l$ is an integer and satisfies that $k^l \geq \frac{||A_{ij}||_1^2}{\epsilon_l^2} \frac{\pi}{2} d^l ||\boldsymbol{X}^{l-1}||_1^2$, $k^0 \geq \frac{||\Omega_{ij}||_1^2}{\epsilon^2} \max_{i,j} ||x_{ij}||^2$. And

$$
||\boldsymbol{X}^l|| = ||\boldsymbol{W}^l \boldsymbol{X}^{l-1}|| \leq ||\boldsymbol{W}^l|| ||\boldsymbol{X}^{l-1}|| \leq ... \leq \prod_{j=1}^{l} ||\boldsymbol{W}^j|| \, ||\boldsymbol{X}^0||.
$$

In order to satisfy that $\tau < \epsilon$, set

$$
\epsilon_i = \frac{\alpha_i \epsilon}{\prod_{j > i} c_j}, \quad \alpha_i = \frac{1}{\bar{\alpha}} (\frac{b_i}{c_i})^{2/3}, \quad \bar{\alpha} = \sum_{i=1}^{L} (\frac{b_i}{c_i})^{2/3}.
$$

Then, by Lemma 5

$$
\tau = \sum_{j \leq L} \epsilon_j \prod_{l=j+1}^{L} c_l = \sum_{j=1}^{L} \alpha_i \epsilon_i = \epsilon.
$$

And

$$ln(N_d(H_X, \tau, |\cdot|_{L+1})) \leq \sum_{l=1}^{L} ln(2d^l d^{l-1} + 1) \cdot \frac{b_l^2}{\epsilon_l^2} \frac{\pi}{2} d^l \cdot ||\boldsymbol{X}^0||^2 \prod_{j<l} c_j^2 + ln(4d^0 d^x) \frac{a_1^2}{\epsilon^2} B^2$$

$$= \sum_{l=1}^{L} ln(2d^l d^{l-1} + 1) \cdot \frac{b_l^2}{\epsilon^2 c_l^2 \alpha_l^2} \frac{\pi}{2} d^l \cdot ||\boldsymbol{X}^0||^2 \prod_{j=1}^{L} c_j^2 + ln(4d^0 d^x) \frac{a_1^2}{\epsilon^2} B^2$$

$$= ln(2W^2 + 1) \frac{\pi}{2\epsilon^2} W ||\boldsymbol{X}^0||^2 \sum_{l=1}^{L} \frac{b_l^2}{c_l^2 \alpha_l^2} \prod_{j=1}^{L} c_j^2 + ln(4W^2) \frac{a_1^2}{\epsilon^2} B^2 \ (d^l \leq W)$$

$$= ln(2W^2 + 1) \frac{\pi}{2\epsilon^2} W ||\boldsymbol{X}^0||^2 (\overline{\alpha})^3 \prod_{j=1}^{L} c_j^2 + ln(4W^2) \frac{a_1^2}{\epsilon^2} B^2 \triangleq \frac{R}{\epsilon^2}.$$

Consider the class of networks $\mathcal{F}$ obtained by affixing the loss $\mathcal{L}(F(x), y)$ and $\mathcal{L}(F(x), y) \leq M$. When $\mathcal{L}$ is fixed, the covering number of the obtained network $\mathcal{F}$ is not larger than the original network.

Then, by Lemma 3

$$R(\frac{\mathcal{F}_{|S}}{M}) \leq \inf_{\alpha>0}(\frac{4\alpha}{\sqrt{n}} + \frac{12}{n} \int_{\alpha}^{\sqrt{n}} \sqrt{lnN_d(\frac{F_\Theta}{M}, \epsilon, ||\cdot||_2)}d\epsilon) = \inf_{\alpha>0}(\frac{4\alpha}{\sqrt{n}} + \frac{12}{n} \int_{\alpha}^{\sqrt{n}} \sqrt{lnN(\frac{H_X}{M}, \frac{\tau}{M}, ||\cdot||_2)}d\epsilon)$$

$$\leq \inf_{\alpha>0}(\frac{4\alpha}{\sqrt{n}} + \frac{12}{n} \int_{\alpha}^{\sqrt{n}} \sqrt{\frac{R}{M^2 \epsilon^2}}d\epsilon) = \inf_{\alpha>0}(\frac{4\alpha}{\sqrt{n}} + ln(\sqrt{n}/\alpha)\frac{12\sqrt{R}}{Mn})$$

$$\leq \frac{4}{n^{3/2}} + ln(n^{3/2})\frac{12\sqrt{R}}{Mn}(\alpha = 1/n)$$

$$= \frac{4}{n^{3/2}} + ln(n^{3/2})\frac{12\sqrt{ln(2W^2+1)\frac{\pi}{2}W||\boldsymbol{X}^0||^2(\overline{\alpha})^3 \prod_{l=1}^{L} c_l^2 + ln(4W^2)a_1^2 B^2}}{Mn}.$$

And the empirical Rademacher complexities of $\frac{\mathcal{F}}{M}$ follows:

$$\hat{R}_S(\frac{\mathcal{F}}{M}) = \underset{\sigma}{E}[\sup_{\frac{f}{M} \in \frac{\mathcal{F}}{M}} \frac{1}{n} \sum_{i=1}^{n} \sigma_i \frac{f(\boldsymbol{x}_i)}{M}] = \frac{1}{M} \underset{\sigma}{E}[\sup_{f \in \mathcal{F}} \frac{1}{n} \sum_{i=1}^{n} \sigma_i f(x_i)] = \frac{1}{M}\hat{R}_S(\mathcal{F}).$$

By Lemma 4

$$\mathbb{E}_{(\boldsymbol{x},\boldsymbol{y}) \sim \mathcal{D}}[(\mathcal{L}(F_\Theta(\boldsymbol{x}), \boldsymbol{y})] \leq \frac{1}{n} \sum_{i=1}^{n} \mathcal{L}(F_\Theta(\boldsymbol{x}_i), \boldsymbol{y}_i) + 2\hat{R}_S(\mathcal{F}) + 3M\sqrt{\frac{ln\frac{2}{\delta}}{2n}}$$

$$\leq \frac{1}{n} \sum_{i=1}^{n} \mathcal{L}(F_\Theta(\boldsymbol{x}_i), \boldsymbol{y}_i) + 3M\sqrt{\frac{ln\frac{2}{\delta}}{2n}} + \frac{8M}{n^{3/2}}$$

$$+ ln(n^{3/2}) \frac{24\sqrt{ln(2W^2+1)\frac{\pi}{2}W||\boldsymbol{X}^0||^2(\overline{\alpha})^3 \prod_{l=1}^{L} c_l^2 + ln(4W^2)a_1^2 B^2}}{n}$$

$$\leq \frac{1}{n} \sum_{i=1}^{n} \mathcal{L}(F_\Theta(\boldsymbol{x}_i), \boldsymbol{y}_i)$$

$$+ \mathcal{O}(\frac{8M}{n^{3/2}} + M\sqrt{\frac{ln1/\delta}{n}} + ln(n)\frac{\sqrt{ln(W)W||\boldsymbol{X}^0||^2 T^2 + ln(W)a_1^2||\boldsymbol{X}||^2}}{n}),$$

where $T^2 = (\overline{\alpha})^3 \prod_{l=1}^{L} c_l^2$. $\qquad\square$

## 2 Initialization

As mentioned in the main paper, we initialize the complex-valued weight matrix as $\boldsymbol{W} = \cos(\boldsymbol{A}) + i\sin(\boldsymbol{A})$. This design ensures CosNet retains the property of non-stationary spectral kernels and

takes the relative distance of data in the complex number domain without increasing the number of parameters. In this section, we further discuss the initialization, including non-stationary and multi-kernel learning.

**Non-stationary ensuring**  Compared with the CSKE module of CosNet, sampling stack the complex-valued spectral kernel mapping in the neural networks cannot ensure that the model retains the non-stationary of the spectral kernel. In this section, we explain that in a two-dimensional case.

For the complex-valued spectral mapping $z = \begin{bmatrix} \cos(u_{11}) + \cos(u'_{11}) \\ \cos(u_{21}) + \cos(u'_{21}) \end{bmatrix} + i \begin{bmatrix} \sin(v_{11}) + \sin(v'_{11}) \\ \sin(v_{21}) + \sin(v'_{21}) \end{bmatrix} \in$ $\mathbb{C}^2$, following the commonly used setting, the weight matrix is defined as $\boldsymbol{W} = \boldsymbol{A} + i\boldsymbol{B}$. The complex-valued transformation with the matrix formula can be defined as:

$$
\Psi(z) = \begin{bmatrix} \boldsymbol{A} & -\boldsymbol{B} \\ \boldsymbol{B} & \boldsymbol{A} \end{bmatrix} * \begin{bmatrix} \Re(z) \\ \Im(z) \end{bmatrix} = \begin{bmatrix} a_{11} & b_{11} \\ a_{12} & b_{12} \\ -b_{11} & a_{11} \\ -b_{12} & a_{12} \end{bmatrix}^{\top} * \begin{bmatrix} \cos(u_{11}) + \cos(u'_{11}) \\ \cos(u_{21}) + \cos(u'_{21}) \\ \sin(v_{11}) + \sin(v'_{11}) \\ \sin(v_{21}) + \sin(v'_{21}) \end{bmatrix}
$$
$$
= \begin{bmatrix} a_{11}(\cos(o_{11}) + \cos(o'_{11})) + a_{12}(\cos(u_{21}) + \cos(u'_{21})) - b_{11}(\sin(v_{11}) + \sin(v'_{11})) - b_{12}(\sin(v_{21}) + \sin(v'_{21})) \\ b_{11}(\cos(u_{11}) + \cos(u'_{11})) + b_{12}(\cos(u_{21}) + \cos(u'_{21})) + a_{11}(\sin(v_{11}) + \sin(v'_{11})) + a_{12}(\sin(v_{21}) + \sin(v'_{21})) \end{bmatrix},
$$

Obviously, for a general non-stationary spectral kernel $k(z, z')$, it cannot be defined as the inner product of two complex-valued mappings, $\Psi(z)$ and $\overline{\Psi(z')}$.

**Multi-kernels learning**  In addition to the mentioned property in the main paper, our initialization enables each unit can be considered a combination of multiple spectral kernels. In this section, we explain that in two-dimensional complex input space.

**Example 1.** *Let*

$$
z = \begin{bmatrix} \cos(u_{11}) + \cos(u'_{11}) \\ \cos(u_{21}) + \cos(u'_{21}) \end{bmatrix} + i \begin{bmatrix} \sin(v_{11}) + \sin(v'_{11}) \\ \sin(v_{21}) + \sin(v'_{21}) \end{bmatrix} \in \mathbb{C}^2
$$

*be a complex-valued vector. The complex-valued weight matrix is defined as $\boldsymbol{W} = \cos(\boldsymbol{A}) + i\sin(\boldsymbol{A})$, where $\boldsymbol{A} = [a_{11}, a_{12}] \in \mathbb{R}^{1 \times 2}$ is a real-valued matrix. The complex-valued mapping can be defined as:*

$$
\Psi(z) = \boldsymbol{W} * z
$$
$$
= (\cos(\boldsymbol{A}) + i\sin(\boldsymbol{A}))
$$
$$
* (\begin{bmatrix} \cos(u_{11}) + \cos(u'_{11}) \\ \cos(u_{21}) + \cos(u'_{21}) \end{bmatrix} + i \begin{bmatrix} \sin(v_{11}) + \sin(v'_{11}) \\ \sin(v_{21}) + \sin(v'_{21})) \end{bmatrix})
$$

*Without loss of generality, we formalize the complex-valued mapping as the following matrix notation:*

$$
\Psi(z) = \begin{bmatrix} \cos(\boldsymbol{A}) & -\sin(\boldsymbol{A}) \\ \sin(\boldsymbol{A}) & \cos(\boldsymbol{A}) \end{bmatrix} * \begin{bmatrix} \Re(z) \\ \Im(z) \end{bmatrix}
$$
$$
= \begin{bmatrix} \cos(a_{11}) & \sin(a_{11}) \\ \cos(a_{12}) & \sin(a_{12}) \\ -\sin(a_{11}) & \cos(a_{11}) \\ -\sin(a_{12}) & \cos(a_{12}) \end{bmatrix}^{\top} * \begin{bmatrix} \cos(u_{11}) + \cos(u'_{11}) \\ \cos(u_{21}) + \cos(u'_{21}) \\ \sin(u_{11}) + \sin(u'_{11}) \\ \sin(u_{21}) + \sin(u'_{21}) \end{bmatrix}
$$
$$
= \begin{bmatrix} \Psi_{a_{11},u_{11},u'_{11}} + \Psi_{a_{12},u_{21},u'_{21}} \\ \Psi'_{a_{11},v_{11},v'_{11}} + \Psi'_{a_{12},v_{21},v'_{21}} \end{bmatrix}
$$

*where $\Psi_{a_{11},u_{11},u'_{11}} = \cos(a_{11} + u_{11}) + \cos(a_{11} + u'_{11})$, $\Psi_{a_{12},u_{21},u'_{21}} = \cos(a_{12} + u_{21}) + \cos(a_{12} + u'_{21})$, $\Psi'_{a_{11},v_{11},v'_{11}} = \sin(a_{11} + v_{11}) + \sin(a_{11} + v'_{11})$, and $\Psi'_{a_{12},v_{21},v'_{21}} = \sin(a_{12} + v_{21}) + \sin(a_{12} + v'_{21})$.*

We can observe that $\Psi_{a_{11},u_{11},u'_{11}}$, $\Psi_{a_{12},u_{21},u'_{21}}$, $\Psi'_{a_{11},v_{11},v'_{11}}$, and $\Psi'_{a_{12},v_{21},v'_{21}}$ can be seen as two separate parts of two different spectral kernel mappings. Hence, the proposed CosNet can be regarded as a linear combination of different kernels (the number of kernels is restricted by the feature numbers), which indicates that our method naturally has a close relation with multi-kernel learning.

# 3  Experiment

In this section, we include more details of the experiment section in the main paper, including the information on the involved datasets (shown in Table 1), the detailed setting for each dataset (shown in Table 2), and extra experiments.

Table 1: The detailed information of the involved dataset. Specifically, the input size denotes the number of time points and features for the time-series classification task and regression task, respectively.

| Dataset | Type | Input size | Train.Data | Test.Data | Class |
|---|---|---|---|---|---|
| FordA | Sensor | 500 | 3601 | 1320 | 2 |
| FordB | Sensor | 500 | 3636 | 810 | 2 |
| PhalangesOutlinesCorrect | Image | 80 | 1800 | 858 | 2 |
| Wine | Spectro | 234 | 57 | 54 | 2 |
| ECG200 | ECG | 96 | 100 | 100 | 2 |
| ECG5000 | ECG | 140 | 500 | 4500 | 5 |
| Herring | Image | 512 | 64 | 64 | 2 |
| Ham | Spectro | 431 | 109 | 105 | 2 |
| ProximalPhalanxOutlineAgeGroup | Image | 80 | 400 | 139 | 6 |
| Earthquakes | Sensor | 512 | 322 | 139 | 2 |
| DistalPhalanxTW | Image | 80 | 400 | 139 | 6 |
| Strawberry | Spectro | 235 | 613 | 370 | 2 |
| power | – | 4 | 7654 | 1914 | – |
| concreat | – | 8 | 824 | 206 | – |
| yacht | – | 6 | 246 | 62 | – |

Table 2: The detailed settings on different datasets. Specifically, Init denotes that the weight matrix is sampled from $\mathcal{N}(0, p)$. Networks denote the unified architecture, where, the first number is the input size, the last number is the class, and the others denote the neuron numbers of the hidden layers.

| | Networks |
|---|---|
| FordA | $500 \times 500 \times 256 \times 64 \times 2$ |
| FordB | $500 \times 500 \times 256 \times 64 \times 2$ |
| PhalangesOutlinesCorrect | $80 \times 80 \times 80 \times 64 \times 2$ |
| Wine | $234 \times 234 \times 128 \times 64 \times 2$ |
| ECG200 | $96 \times 96 \times 96 \times 32 \times 2$ |
| ECG5000 | $140 \times 140 \times 64 \times 32 \times 5$ |
| Herring | $512 \times 512 \times 128 \times 64 \times 2$ |
| Ham | $431 \times 431 \times 128 \times 64 \times 2$ |
| ProximalPhalanxOutlineAgeGroup | $80 \times 80 \times 80 \times 32 \times 3$ |
| Earthquakes | $512 \times 512 \times 128 \times 32 \times 2$ |
| DistalPhalanxTW | $80 \times 80 \times 80 \times 32 \times 6$ |
| Strawberry | $235 \times 235 \times 128 \times 32 \times 2$ |
| power | $4 \times 4 \times 4 \times 4 \times 1$ |
| concreat | $8 \times 8 \times 8 \times 4 \times 1$ |
| yacht | $6 \times 6 \times 6 \times 3 \times 1$ |

**Image classification**   Addition to the time-squential data, complex-valued representation is commonly used in the image processing. The phase describes objects in an image in terms of edges, shapes and their orientation. To explore the capability of CosNet on image-related tasks, we extend CosNet to the convolutional neural networks (CNNs), namely complex-valued spectral convolutional networks (CosCNet). Similar with CosNet, CosCNet also include two modules, including complex-valued representation learning (CRL) module and complex-valued convolutional (CC) module. The CRL module is used to transform the image in the real number domain to the complex number domain, and the CC module is used to explore the inherently complex-valued representation and further explore the detailed information of edges and shape.

Table 3: Classification accuracy (%) under different hyper-parameters. The best results are highlighted in **bold**.

| lr | init (p) | SRFF | DSKN | $DCN^1$ | $DCN^2$ | ASKL | CosNet |
|---|---|---|---|---|---|---|---|
| 0.1 | 1 | 64 | 60.85 | **90.30** | 83.15 | 61.00 | 86.05 |
| 0.1 | 0.1 | 85.55 | 61.50 | **90.30** | 83.15 | 80.20 | 85.85 |
| 0.1 | 0.01 | 78.25 | 62.80 | **90.30** | 83.15 | 74.10 | 85.05 |
| 0.01 | 1 | 52.60 | 64.00 | 88.10 | 84.05 | 75.00 | **90.05** |
| 0.01 | 0.1 | 85.40 | 67.95 | 88.10 | 84.05 | 89.75 | **91.30** |
| 0.01 | 0.01 | 73.40 | 77.80 | 88.10 | 84.05 | 87.53 | **90.10** |
| 0.001 | 1 | 50.85 | 62.75 | 80.35 | 79.25 | 72.55 | **89.25** |
| 0.001 | 0.1 | 83.50 | 73.00 | 80.35 | 79.25 | **90.90** | 90.25 |
| 0.001 | 0.01 | 64.00 | 84.40 | 80.35 | 79.25 | 88.90 | **90.45** |

Specifically, the CRL module is defined as :

$$\Phi(\boldsymbol{x}) = \sqrt{\frac{1}{4M}} \Big[ (\cos(\boldsymbol{\Omega} * \boldsymbol{x}) + \cos(\boldsymbol{\Omega}' * \boldsymbol{x})) + i(\sin(\boldsymbol{\Omega} * \boldsymbol{x}) + \sin(\boldsymbol{\Omega}' * \boldsymbol{x})) \Big],$$

and the convolution operation of CC module with the matrix notation is defined as:

$$\begin{bmatrix} \Re(\Psi(\boldsymbol{h})) \\ \Im(\Psi(\boldsymbol{h})) \end{bmatrix} = \begin{bmatrix} \cos(\boldsymbol{A}) & -\sin(\boldsymbol{A}) \\ \sin(\boldsymbol{A}) & \cos(\boldsymbol{A}) \end{bmatrix} * \sqrt{\frac{1}{4M}} \begin{bmatrix} \cos(\boldsymbol{\Omega} * \boldsymbol{x}) + \cos(\boldsymbol{\Omega}' * \boldsymbol{x}) \\ \sin(\boldsymbol{\Omega} * \boldsymbol{x}) + \sin(\boldsymbol{\Omega}' * \boldsymbol{x}) \end{bmatrix},$$

where, $\boldsymbol{\Omega}$, $\boldsymbol{\Omega}'$, and $\boldsymbol{A}$ are filters. Moreover, the CosCNet with $l$ layers is defined as:

$$CosCNet(\boldsymbol{x}) = \Psi^{l-1}(\ldots \Psi^1(\Phi^1(\boldsymbol{x}))).$$

**Generalizaion of CosNet** Furthermore, to evaluate the generalization of our CosNet, we explore the influence of varying learning rates and distribution of weight matrics on the result based on ECG200 dataset. The results are shown in Table 3 The results show the superior performance and stability of our CosNet.