# OpenReview forum: "CosNet: A Generalized Spectral Kernel Network"
_NeurIPS.cc/2023/Conference — NeurIPS 2023 poster_

### Official Review · Reviewer_uy3z · 2023-06-11

**Soundness:** 3 good
**Presentation:** 1 poor
**Contribution:** 3 good
**Rating:** 3
**Confidence:** 3

**Summary:**

The authors propose a complex valued neural architecture, composed of two modules: a Spectral kernel mapping generalization module and a Complex-valued spectral kernel embedding module.
They provide a generalization error bound for their model. In addition they propose a novel initialization scheme and provide experimental results on several datasets and learning tasks.

**Strengths:**

The proposed approach seem to be novel, and the experimental results are promising.

**Weaknesses:**

In my view, the main weakness of the manuscript is the presentation. While the approach itself seems sound, I find the presentation too poor for a conference like Neurips. Therefore I encourage the authors to re-write the manuscript and make sure it reads much better.

**Questions:**

No

---

> ### Author Rebuttal · Authors · 2023-08-09
>
> **Q: In my view, the main weakness of the manuscript is the presentation. While the approach itself seems sound, I find the presentation too poor for a conference like Neurips. Therefore I encourage the authors to re-write the manuscript and make sure it reads much better.**
>
> **Response:**
> We sincerely appreciate your time and effort in reviewing our manuscript. In response to your comment regarding the presentation of our manuscript, we would like to bring to your attention that reviewers gGnJ and BAqh have both commended the contributions of our CosNet. They found our methodology and approach to be well-explained and insightful. This suggests that our efforts to enhance the clarity and presentation of our work have been positively received by these reviewers.
> Nonetheless, we understand the importance of ensuring the highest level of clarity and coherence throughout the manuscript. We value your more constructive feedback and are dedicated to refining our manuscript based on your comments. We eagerly anticipate the opportunity to incorporate more of your insights into our revision process.

---

> > ### Comment · Reviewer_uy3z · 2023-08-15
> > **Acknowledging read of the rebuttal**
> >
> > I thank the authors for their rebuttal and effort.
> > I will leave my rating as-is.
> > Yet, indeed it seems that two of the other reviewers find the manuscript clearer than I did. I will leave it to the AC to take it further.

---

> > > ### Author Response · Authors · 2023-08-15
> > > **Thanks!**
> > >
> > > Thanks for your reply.

---

### Official Review · Reviewer_BAqh · 2023-06-28

**Soundness:** 3 good
**Presentation:** 3 good
**Contribution:** 3 good
**Rating:** 7
**Confidence:** 4

**Summary:**

The paper proposes a new framework called Complex-valued spectral kernel network (CosNet) that generalizes the spectral kernel to include complex-valued representation. The proposed framework improves the representational capability of the spectral kernel and outperforms existing kernel methods and complex-valued neural networks. An initialization scheme for the complex-valued weight matrix is proposed, which ensures that CosNet retains the property of non-stationary spectral kernels and takes the relative distance of data in the complex number domain without increasing the number of parameters. The paper provides the lower generalization bound of CosNet than the real-valued non-stationary spectral kernel. The experiments demonstrate that CosNet performs better than the mainstream kernel methods and complex-valued neural networks in time-sequential data analysis.

**Strengths:**

1. The paper proposes a new framework (CosNet) and provides a theoretical analysis of it.
2. The experiments demonstrate that CosNet outperforms existing kernel methods in time-series data analysis, which shows the practical significance of the proposed framework.
3. The writing of the paper is good.

**Weaknesses:**

1. The experiments presented in the paper lack sufficient evidence to convincingly demonstrate the properties of the model.
2. The dataset used in the experiments is limited in diversity, which raises concerns about the generalizability of the findings.
3. The initialization of CosNet's parameters varies across different datasets, and a unified initialization method is needed for consistency and reproducibility.

Typo: In line 168, the imaginary part should also be multiplied by the weight $1/\sqrt{\frac{1}{4M}}$.

**Questions:**

1. The paper presents experiments comparing the accuracy of CosNet to other methods with similar network size on the same dataset, highlighting the capacity gain from introducing the imaginary part. However, I believe that accuracy alone may not fully demonstrate the improvement of the model's representation capacity. Could the authors provide more evidence of CosNet's capacity gain from other aspects, such as compression and recovery of more complex datasets?
2. The experiments in the paper adopt time-series datasets with low complexity. Could the authors provide more results that demonstrate the performance of CosNet on tasks such as image encoding and decoding or the signal processing in real-world scenarios, since these datasets may also be suitable cases for complex-valued spectral kernel modeling.
3. In Figure 2, the authors only present the visualization results of two other methods for predicting and displaying the original time series curve. Could the authors provide additional visualizations of the performance of other comparable methods for comparison? This would provide a more comprehensive and fair comparison of the proposed method against existing ones.

---

> ### Author Rebuttal · Authors · 2023-08-09
>
> **Q1: The initialization of CosNet's parameters varies across different datasets, and a unified initialization method is needed for consistency and reproducibility.**
>
> **Response:** To ensure the reproducibility of our experimental findings, we unify the hyper-parameters, and the partial updated results (under the same learning rate (0.01), initialization (p = 0.01), and layer numbers (5)) are reported in the following Table. The new results show that our CosNet performs better than the baseline methods, and all the related results will be reported in the revised main paper.
> | Dataset                        | SRFF   | DSKN   | \(DCN^1\) | \(DCN^2\) | ASKL   | CosNet  |
> |-------------------------------|--------|--------|----------|----------|--------|---------|
> | FordB                         | 68.99  | 69.81  | 69.68  | 50.17  | 64.20  | **71.73** |
> | Wine                          | 77.22  | 76.48  | 83.06  | 80.00  | 67.41  | **85.46** |
> | ECG200                        | 73.40  | 77.80  | 89.80  | 89.85  | 87.53  | **90.10** |
> | ECG5000                       | 91.98  | 91.14  | 94.11  | 93.50   | 92.75  | **93.70**  |
> | Herring                       | 57.73  | 56.64  | 65.23  | 58.13  | 59.52  | **65.39** |
>
> **Q2: The experiments in the paper adopt time-series datasets with low complexity. Could the authors provide more results that demonstrate the performance of CosNet on tasks such as image encoding and decoding or the signal processing in real-world scenarios, since these datasets may also be suitable cases for complex-valued spectral kernel modeling.**
>
> **Response:** On the one hand, we have included the automatic modulation (AM) classification task using the real-world signal dataset RML2016.10a [1], a classical complex-valued signal dataset. We conduct a comparative analysis between our proposed method and baseline approaches across varying signal-to-noise ratios (SNRs) ranges. On the other hand, we expand the application of CosNet to convolutional networks for image classification tasks on the FashionMNIST and CIFAR-10 datasets.  All the results show the effectiveness of our CosNet. Detailed results are provided in the following table for a comprehensive overview and will be incorporated into the revised manuscript.
> | Tasks               | Datasets     | \(DCN^1\) | \(DCN^2\) | CosNet   |
> |---------------------|--------------|----------|----------|----------|
> | Image classification| FashionMNIST | 87.02    | 84.44    | **88.33**|
> |                     | CIFAR-10     | 64.32    | 52.39    | **66.51**|
>
> | Tasks              | SNRs range | \(DCN^1\) | \(DCN^2\) | CosNet   |
> |--------------------|------------|----------|----------|----------|
> | AM classification  | 10-18      | 81.14  | 79.98 | **81.89**|
> |                    | 0-8  | 79.27 | 77.45 | **79.70**|
>
> [1] O'shea T J, West N. Radio machine learning dataset generation with gnu radio[C]//Proceedings of the GNU Radio Conference. 2016, 1(1).
>
> **Q3: The paper presents experiments comparing the accuracy of CosNet to other methods with similar network size on the same dataset, highlighting the capacity gain from introducing the imaginary part. However, I believe that accuracy alone may not fully demonstrate the improvement of the model's representation capacity. Could the authors provide more evidence of CosNet's capacity gain from other aspects, such as compression and recovery of more complex datasets?**
>
> **Response:**  To further explore the representation of our CosNet, we conduct more complex task utilizing the FashionMNIST and CIFAR-10 datasets. Concretely, we first extract the implicit features through various models, and then we conduct the clustering task based on these extracted features. In this task, Normalized Mutual Information (NMI) and Rand Index (RI) are used as the assessment metrics. Our CosNet achieves **5.6\%** NMI improvement (86.04\% $\rightarrow $90.86\%), **1.19\%** RI improvement (97.15\% $\rightarrow$ 98.13\%) on FMNIST dataset, and **15.82\%** NMI improvement (57.14\% $\rightarrow$ 66.18\%), **2.5\%** RI improvement (87.78\% $\rightarrow$ 89.97\%) on CIFARI-10 dataset.  The results show that our CosNet has a greater representation capbility than other complex-valued convolutional networks. Detailed results are provided in the following table. Further, we will include more compression and recovery tasks in the revised manuscript.
> | Dataset  | Metrics | \(DCN^1\) | \(DCN^2\) | CosNet   |
> |----------|---------|----------|----------|----------|
> | FMNIST   | NMI     | 86.04\%   | 81.31\%   | **90.86\%**|
> |          | RI      | 97.15\%   | 95.97\%   | **98.31\%**|
> | CIFAR-10 | NMI     | 57.14\%   | 41.69\%   | **66.18\%**|
> |          | RI      | 87.78\%   | 84.07\%   | **89.97\%**|
>
> **Q4: In Figure 2, the authors only present the visualization results of two other methods for predicting and displaying the original time series curve. Could the authors provide additional visualizations of the performance of other comparable methods for comparison? This would provide a more comprehensive and fair comparison of the proposed method against existing ones.**
>
> **Response:**  Figure 2 is designed to illustrate the effective capture of inherently complex-valued representations by the first layer of our CosNet, seamlessly feeding them into the subsequent complx-valued networks (*i.e.* the CSKE module). To validate this statement, we utilize DSKN and FT as the baseline methods, driven by two primary reasons. Firstly, Fourier transform is extensively utilized to map real-valued data into complex-valued representations, which serve as inputs for complex-valued networks. Second, DSKN encompasses most existing approaches, where the imaginary part is padded with zeros. While DSKN ignores the imaginary part that is learned from the real-valued data, the other methods directly consider the raw real-valued data and zeros as the real and imaginary parts. As a result, we select these two methods for comparison purposes.

---

### Official Review · Reviewer_YSFk · 2023-07-06

**Soundness:** 2 fair
**Presentation:** 1 poor
**Contribution:** 3 good
**Rating:** 6
**Confidence:** 3

**Summary:**

The paper aims to extend the reach of kenel-based inference for time series by using a hilbert space over a complex field rather than working over the reals, by not discarding the phase component of the implied spectral representation of the kernels, which apparently is a common strategy. This method makes it feasible to define a broader class of kernels than the obvious (.e.g stationary)

**Strengths:**

The paper seems to propose a novel way of characterising flexible covariance kernels such that they are PSD, and some neural networks that exploit this

**Weaknesses:**

This paper is hard to read. The main claims are difficult to extract and thus their correctness is hard to verify.
I could be persuaded that I have underestimated the soundness of the results --- they could in fact be amazing --- but I have already blown my limited time budget trying to understand what is going on, because basic framing information is missing.
I presume that some of this could be deduced by inspecting the references introduced in section 2, but I don't have time for that in the setting of urgent conf reviews. the paper needs to be self-contained, even if it we defer rigorous proof to the appendices.

My certainty rating reflects my relatively high confidence that this paper needs a rewrite. I do not have a high confidence that the idea itself is flawed, and would welcome clarifications that showed me what the idea actually is, because it might be great. I currently do no understand it, because the presentation is confusing enough that it will take more than the time I have to "reverse engineer" it.

**Questions:**

There are a lot of words about the usefulness of the complex-values covariance kernel which I think I can interpret but I do not actually know for sure, because the actual inference problem is not clearly set up. What is it? What is the baseline?

I think my confusion starts in l122. Everything up to this point was fine, but now we have deployed the machinery of a complex-valued spectrum in  kernel definitions, without actually explaining what is different. there should be certain things that are better for a complex kernel from the Yaglom theorem. For one, a kernel so defined does needs to be stationary, as opposed to a Bochner-theorem style kernel, which I assume is  the main point, and very cool and indeed the authors mention this right in the abstract. So I like this! I've looked at Yaglom's theorem before and though tit would be great to make it tractable to use but I didn't see a way. Indeed this should be a very general kernel, and should generalise other kernel classes too (e.g. dot-product kernels). So the idea sounds promising.
Even here though, I'm a little confused; OK, so we are using this kernel not as, say, a covariance kernel of a Gaussian process, but rather to directly define a similarity between data points for optimal interpolation, as far as I can tell, which is fine, but can you say more about that actual implied network structure? Do we keep around the training data so we can measure the kernel-similarity to other training points, or are we happy to use it as "just another" nonlinear NN layer. In which case, what does this layer do that an MLP does *not* do?

in eq (10) we learn that the kernel is characterised by a finite list  $\Omega$ of frequencies, right? Is it correct that this means that we are restricting our kernel spectral "density" $\left(\boldsymbol{\omega}, \boldsymbol{\omega}^{\prime}\right)$ to be not a density as such but  but rather to be a collection of dirac deltas in the spectral space? I suspect we need to say so, in that case.

l147/eq(8) the $M$ suddenly appeared and it looks important but is not explored. This is the number of MC samples we take to actually approximate the spectral kernel integral. So... when do we evaluate this integral? Is it inside the training loop? How do we choose $M$? is it robust against different choices of $M$? Should we not see $M$ pop up in evaluating the computational cost of this method.

Can we put a simple function map notation description of *every operator and function*, e.g. $\Phi_\ell: \mathbb{C}^{d_{\ell}}\to\mathbb{C}^{d_{\ell+1}}$?  Generally, for most functions in this paper I'm confused what the are mapping from and to. I have so many questions here that I cannot list them all. Here are some examples about function domains and ranges:

1. Are we permitting the values of the kernel outputs in the intermediate layers to be complex?
2. which values are permitted to be vectors and which scalars? I gather the inputs to the network are vectors (?) but the test examples seem to be all time series with a scalar index (?) In section 4,1a the experiments seem to be about estimating complex valued time series, and in section 4.1b it is classifying stuff based on the implied feature mapping. Is the model more general than this?

If the authors wish to remain very general, fine, but if so, could we have a running example that makes it clear?

Since we also know that NNs exist which do not parameterise their output in terms of parameters of kernels, but directly in terms of _weights_, could we say anything about the relative representation power compared to that?
There is Theorem 1, which gives us a statistical learning theory result in terms of covering numbers, which is possibly supposed to help us. Perhaps the problem here is my own ignorance, but what is the equivalent results for the baseline that the authors hope to surpass? Can you quote an equivalent theorem for a baseline? Should I know one? Or is the intent here to show us how to trade off between allocating weights to _including more frequencies in a given kernel layer_ versus _adding more layers_? Can you spell out the implications of this theorem in terms of "wide versus deep complex kernel networks" and also "whether complex kernel networks are better than MLPs"?

Related: Why do we need to stack layers at all, rather than simply learning a big kernel?

4.1: There is a neural network here. I support the authors not wasting space with excessive details about Adagrad learning rates etc, but I need just a little more information to know what is happening. What is the training procedure? If I am used to ERM for training an NN, do I need to worry about some alternative methods for these exotic kernel networks, or is it the same? If it is the same, why am I thinking about the kernels directly rather than just learning an MLP? remember, since the network has been non-specific throughout the paper, this is my example to learn what actual inputs and outputs this network can predict upon.

**Limitations:**

I don't know. I have a hard time deducing exactly the domain of applicability of the paper from the presentation here.

---

> ### Author Rebuttal · Authors · 2023-08-09
>
> Thanks for your comments! We will provide a point-to point response in the rebuttal.
>
> **1 Notations** In this paper, the matrices, vectors and scalars are denoted
> by bold capital letters (*e.g.* $\pmb{X}$), bold lower-case letters (*e.g.* $\pmb{x}$) and lower-case letters (*e.g.* $x$), respectively. In addition, for each equation or function, we will includ its domain and range in the revised manuscript.
>
> **2 Network architecture and experiment setting:**  In the inference procedure, as exemplified by time series classification task, the input is a time series (*i.e.* vector) with a scalar at each time point. The output is the implied feature (*i.e.* vector), which is used to conduct the classification task. Concretely, the operation in the first layer is defined as $\Phi:\mathbb{R}^{d^x}\rightarrow \mathbb{C}^{d^x}$, where $d^x$ denotes the dimension of the data. Via $\Phi$ in the first layer, the data result in complex-valued representations, which are fed into the CSKE module starting from the second layer. The operation of $l^{th}$ layer is defined as $\Psi^l: \mathbb{C}^{d^l}\rightarrow \mathbb{C}^{d^{l+1}}$, where $d^l$ denotes the number of hidden complex-valued neuron. After CSKE module, we obtain the implied complex-valued feathers. Moreover, these implied complex-valued features are condensed into vector form by the operation $\mathbb{C}^{d^L}\rightarrow \mathbb{R}^{2d^L}$, which concatenate real and imaginary parts, to conduct the classification task.
>
> In the experiment, the learning rate, epoch and layer number are set as 0.01, 500 and 5, respectively. The batch\_size is equal to number of samples and the width of networks in each dataset depends on the length of time series. The initialized weight matrices are sampled from the normal distribution $\mathcal{N}(0, 0.01)$. The detalied information will be shown in the updated paper.
>
> **3 Kernel or nonlinear NN layer:**  Note that, in CosNet,  the kernel is defined using explicit kernel mapping rather than the covariance matrix. Specifically, Yaglom's theorem establishes a connection between a kernel and its spectral density. Basen on Monte Carlo random sampling, we can approximate the kernel with a explicit kernel mapping within eq(8). Our CosNet is constructed via stacking the explicit kernel mapping with multiple layers. Notably, in eq(8), we do not need to caculate the integral in the training process. Here, $M$ is the number of MC samples, also referred to as the count of frequencies, and it equal to the numbers of feature in this paper. Moreover, in order to explore the influence of $M$ on the result, we conduct the experiment based on our CosNet, and $M$ is set in the range $[\frac{d^x}{4}, \frac{d^x}{2}, d^x, 2d^x, 4d^x]$. The result shows that while the value of $M$ does exert some influence on the results, the effect is relatively minor. Please see TABLE I in the added pdf file for more detailed results, and  it will be shown in the revised paper.
>
> **4 Complex-valued kernel network versus MLPs:**  For the framework generality, our CosNet demonstrates the capability to analyze not only complex-valued data but also real-valued data that inherently include complex-valued information. In contrast, MLPs confined to analyzing solely real-valued data. Theoretically, our CosNet has greater representation ability compared to MLPs. Concretely, we bound the covering number of different layers in CosNet. Covering numbers also serves as an indicator of models’ representation ability, where the larger the covering number the greater the representation ability, but the more difficult it is to get the optimal solution. In Theorem 1, the covering number of each layer is bounded by $(2d^ld^{l-1})^k$ and $(4d^ld^{l-1})^k$ in MLPs and real-valued non-stationary spectral kernel networks, respectively. This provides insight into the comparison: 1) Compared with MLPs, our CosNet has greater representation ability with the bound of $(4d^ld^{l-1})^k$ covering numbers in the first layer; 2) Compared with real-valued spectral kernel networks, our CosNet makes it easier to find the optimal solution with the bound of $(2d^ld^{l-1})^k$ covering numbers from second layer. Therefore, our CosNet combines the advantages of kernel networks and MLPs, which has stronger characterization ability and is easier to find the optimal solution. We will show more analysis in the revised paper.
>
> **5 Why do we need to stack layers at all, rather than simply learning a big kernel?** It is important to note that traditional kernel methods are confined to learning a single layer of nonlinear features, potentially constraining their representational capacity. Inspired by neural networks which learn multi-layer hierarchical representations, deep kernel (stacked kernel) is developed to learn hierarchy within Reproducing Kernel Hilbert Space, yielding a cascade of nonlinear features. Therefore, stacked kernel combines the advantags of kernel and neural networks. For a more comprehensive understanding, we refer interested readers to the detailed explanations provided in the reference titled 'Stacked Kernel Network'. Moreover, we perform a comparative analysis using ECG200 dataset between stacked kernel networks and their corresponding big kernels with 1024 Monte Carlo samples. The result show that the stacked kernel performs better than a big kernel with single layer. Please see TABLE II in the added pdf file for more detailed results, and it will be shown in the revised manuscript.
>
> **6 in eq (10) we learn that the kernel is characterised by a finite list $\pmb{\Omega}$
>  of frequencies, right? Is it correct that this means that we are restricting our kernel spectral "density" $(\pmb{\omega}, \pmb{\omega}')$
>  to be not a density as such but but rather to be a collection of dirac deltas in the spectral space? I suspect we need to say so, in that case.**  Yes, you are right! We will statement this point in the revised manuscript.

---

> > ### Comment · Reviewer_YSFk · 2023-08-15
> >
> > Thank you, this is very helpful. Your revised explanation has substantially improved my understanding of the paper and my estimation of the significance of your results. I will revise my rating accordingly.

---

> > > ### Author Response · Authors · 2023-08-15
> > > **Thanks!**
> > >
> > > We appreciate your reply and are delighted that our explanation can help you understand our approach. Your comments are helpful for improving our work, we will include more details in the revised manuscript. If there are any other questions, we would discuss them in a timely manner.

---

> > > > ### Comment · Reviewer_YSFk · 2023-08-20
> > > >
> > > > I have revised my review to a "weak accept" on the basis that I now understand the work of the authors better and agree that it is useful. My qualm is that the revisions required to communicate this in the paper are substantial

---

> > > > > ### Author Response · Authors · 2023-08-20
> > > > >
> > > > > We appreciate your recognition again, and we will include more details in the revised version based on your comments.

---

### Official Review · Reviewer_gGnJ · 2023-07-10

**Soundness:** 3 good
**Presentation:** 3 good
**Contribution:** 3 good
**Rating:** 6
**Confidence:** 4

**Summary:**

This paper mainly focuses on the issue that spectral kernel-based methods often eliminate the imaginary part when analyzing the characteristics of time-sequential data. This limits the representation capability of the spectral kernel. To address this issue, the authors propose a complex-valued spectral kernel network to take both the real and imaginary parts into account. The proposal mainly consists of two parts - the SKMG module recovers the complex-valued representation for the real-valued data and the CSKE module combines the complex-valued spectral kernels and neural networks. Theoretical and empirical results show the proposed method achieves state-of-the-art performance.

**Strengths:**

1. The proposed method is well-motivated. Involving the imaginary part is critical for preserving the amplitude and phase information for data and improving the representational capability of spectral kernel networks.
2. The proposed approach is sound. The complex-valued representation for the real-valued data is recovered by the SKMG module and complex-valued spectral kernels are combined with neural networks via the CSKE module. The two parts are well integrated.
3. Theoretical analysis and experimental evaluation are provided to show the state-of-the-art performance of the proposal.

**Weaknesses:**

1. In Section 3.3, the authors define the complex-valued weight matrix by Equation 11. But is unclear why this design ensures that the sub-network containing the first layer to arbitrary l-th layer is seen as a spectral kernel.
2. The experimental results appear to be dependent on the choices of hyperparameters. Performance of CosNet with different learning rates, initializations and layer numbers should be provided.

**Questions:**

See weakness.

**Limitations:**

Yes, the authors have addressed the limitations.

---

> ### Author Rebuttal · Authors · 2023-08-09
>
> **Q1: In Section 3.3, the authors define the complex-valued weight matrix by Equation 11. But is unclear why this design ensures that the sub-network containing the first layer to arbitrary l-th layer is seen as a spectral kernel.**
>
> **Response:**
> Thanks for this valuable comment. For our CosNet, the first layer (*i.e.*} SKMG module) is constructed from the Yaglom’s theorem,  naturally resulting in a spectral kernel. We ensure the subnetwork from the first layer to the $l$-th layer is a spectral kernel by elaborating the complex-valued weight matrix and the feed-forward procedure which is explained in Equation (12). Upon expanding Equation (12),  it becomes apparent that the sub-network containing the first layer to arbitrary $l$-th layer is seen as a spectral kernel with the defined complex-valued weight. To enhance clarity, we present an illustrative example in $\mathbb{C}^2$ to show more details in the below, and a more detailed explanation will be added to the revised manuscript.
>
> **Example:**
>
> Let
> $$
> \pmb{z} = \begin{bmatrix}
>     \cos(u_{11}) + \cos(u_{11}') \\\\
>     \cos(u_{21}) + \cos(u_{21}')
> \end{bmatrix} + i \begin{bmatrix}
>     \sin(v_{11} + \sin(v_{11}') \\\\
>     \sin(v_{21}) + \sin(v_{21}')
> \end{bmatrix} \in \mathbb{C}^2
> $$
> be the output of first layer. The complex-valued weight matrix is defined as $\pmb{W}=\cos(\pmb{A}) + i\sin(\pmb{A})$, where $\pmb{A}=[a_{11}, a_{12}] \in \mathbb{R}^{1 \times 2}$ is a real-valued matrix. The complex-valued mapping is defined as:
>
> $$
>    \Psi(\pmb{z}) = \pmb{W}*\pmb{z}
>    =(\cos(\pmb{A})+i\sin(\pmb{A})) * (\begin{bmatrix}
>      \cos(u_{11}) + \cos(u_{11}')  \\\\
>      \cos(u_{21}) + \cos(u_{21'})
> \end{bmatrix} + i\begin{bmatrix}
>      \sin(v_{11}) + \sin(v_{11}')  \\\\
>      \sin(v_{21}) + \sin(v_{21}'))
>      \end{bmatrix})
> $$
> We rewrite the complex-valued mapping as the following matrix notation:
>
> $$
> \Psi(\pmb{z})
> =\begin{bmatrix}
>      \cos(\pmb{A}) & -\sin(\pmb{A}) \\\\
>      \sin(\pmb{A}) & \cos(\pmb{A})
> \end{bmatrix} * \begin{bmatrix}
>      \Re(\pmb{z}) \\\\
>      \Im(\pmb{z})
> \end{bmatrix}
> =\begin{bmatrix}
>      \cos(a_{11}) & \sin(a_{11}) \\\\
>      \cos(a_{12}) & \sin(a_{12}) \\\\
>      -\sin(a_{11}) & \cos(a_{11}) \\\\
>      -\sin(a_{12}) & \cos(a_{12})
> \end{bmatrix}^\top *
> \begin{bmatrix}
>      \cos(u_{11}) + \cos(u_{11}')  \\\\
>      \cos(u_{21}) + \cos(u_{21}')  \\\\
>      \sin(u_{11}) + \sin(u_{11}') \\\\
>      \sin(u_{21}) + sin(u_{21}')
> \end{bmatrix}\\
> = \begin{bmatrix}
>     \Psi_{a_{11},u_{11},u_{11}'} + \Psi_{a_{12},u_{21},u_{21}'} \\\\
>     \Psi_{a_{11},v_{11},v_{11}'}' +
>     \Psi_{a_{12},v_{21},v_{21}'}'
> \end{bmatrix}
> $$
> where $\Psi_{a_{11},u_{11},u_{11}'}=\cos(a_{11}+u_{11})+\cos(a_{11}+u_{11}')$, $\Psi_{a_{12},u_{21},u_{21}'}=\cos(a_{12}+u_{21})+\cos(a_{12}+u_{21}')$, $\Psi_{a_{11},v_{11},v_{11}'}'= \sin(a_{11}+v_{11})+\sin(a_{11}+v_{11}')$, and $\Psi_{a_{12},v_{21},v_{21}'}'= \sin(a_{12}+v_{21})+\sin(a_{12}+v_{21}')$.
>
> We can observe that $\Psi_{a_{11},u_{11},u_{11}'}$, $\Psi_{a_{12},u_{21},u_{21}'}$, $\Psi_{a_{11},v_{11},v_{11}'}'$, and $\Psi_{a_{12},v_{21},v_{21}'}'$ can be seen as two separate spectral kernel. Hence, the sub-network containing the first layer to arbitrary $l$-th layer also is a spectral kernel.
>
>
>
>
> **Q2: The experimental results appear to be dependent on the choices of hyperparameters. Performance of CosNet with different learning rates, initializations and layer numbers should be provided.**
>
> **Response:**
> Thanks for this great comment. To ensure the reproducibility of our experimental findings, we unify the hyper-parameters, and the partial updated results (under the same learning rate (0.01), initialization (p = 0.01), and layer numbers (5)) are reported in the following Table. All the related reults will be reported in the revised manuscript.
> | Dataset                      | SRFF   | DSKN   | \(DCN^1\) | \(DCN^2\) | ASKL   | CosNet |
> |-----------------------------|--------|--------|----------|----------|--------|--------|
> | FordB                       | 68.99  | 69.81  | 69.68    | 50.17    | 64.20  | **71.73**  |
> | Wine                        | 77.22  | 76.48  | 83.06    | 80.00    | 67.41  | **85.46**  |
> | ECG200                      | 73.40  | 77.80  | 89.80    | 89.85    | 87.53  | **90.10**  |
> | ECG5000                     | 91.98  | 91.14  | 94.11    | 93.50     | 92.75  | **93.70**   |
> | Herring                     | 57.73  | 56.64  | 65.23    | 58.13    | 59.52  | **65.39**  |
>
> Furthermore, to evaluate the generalization of our CosNet, we explore the influence of varying hyper-parameters on the result based on ECG200 dataset. The result shows the superior performance and stability of our CosNet. Please see the results in the following table for more details, and we will include  these results  and more analysis in the revised manuscript.
>
> | lr     | init (p) | SRFF   | DSKN   | \(DCN^1\) | \(DCN^2\) | ASKL   | CosNet  |
> |--------|----------|--------|--------|----------|----------|--------|---------|
> | 0.1    | 1        | 64     | 60.85  | **90.30**| 83.15    | 61.00  | 86.05   |
> | 0.1    | 0.1      | 85.55  | 61.50  | **90.30**| 83.15    | 80.20  | 85.85   |
> | 0.1    | 0.01     | 78.25  | 62.80  | **90.30**| 83.15    | 74.10  | 85.05   |
> | 0.01   | 1        | 52.60  | 64.00  | 88.10    | 84.05    | 75.00  | **90.05**|
> | 0.01   | 0.1      | 85.40  | 67.95  | 88.10    | 84.05    | 89.75  | **91.30**|
> | 0.01   | 0.01     | 73.40  | 77.80  | 88.10    | 84.05    | 87.53  | **90.10**|
> | 0.001  | 1        | 50.85  | 62.75  | 80.35    | 79.25    | 72.55  | **89.25**|
> | 0.001  | 0.1      | 83.50  | 73.00  | 80.35    | 79.25    | **90.90**| 90.25   |
> | 0.001  | 0.01     | 64.00  | 84.40  | 80.35    | 79.25    | 88.90  | **90.45**|

---

> > ### Comment · Reviewer_gGnJ · 2023-08-15
> > **Thank you for the response**
> >
> > Thank you for the detailed response. It has well improved my understanding of the paper. I am keeping my positive score unchanged for now and no more questions at this time.

---

> > > ### Author Response · Authors · 2023-08-15
> > > **Thanks!**
> > >
> > > We appreciate your recognition and are delighted that our explanation can help you understand our work. We will include more details in the revised manuscript based on your comments.

---

### Author Rebuttal · Authors · 2023-08-09

 We include additional experimental result in the pdf.

---

### Decision · Program_Chairs · 2023-09-21

**Decision:**

Accept (poster)

**Comment:**

This paper explores a key challenge in spectral kernel-based methods: the elimination of the imaginary part, which limits the kernel's representation when analyzing time-sequential data. To solve this problem, the authors propose a complex-valued spectral kernel network. The experiments demonstrate that CosNet outperforms existing methods. During the discussion, the authors provided a good rebuttal. They presented some preliminary results that addressed the reviewers' concerns. However, one reviewer still remains concerned about the presentation of this manuscript, while the other reviewers have not identified that issue.
Most parts of this paper are well presented, and the authors should further improve this manuscript based on the rebuttal during the discussion period.
Overall, I recommend accepting this paper.